# Chlorpromazine eliminates acute myeloid leukemia cells by perturbing subcellular localization of FLT3-ITD and KIT-D816V

Shinya Rai [1,6], Hirokazu Tanaka [1,6 ✉], Mai Suzuki[2], J. Luis Espinoza [1], Takahiro Kumode[1], Akira Tanimura[3], Takafumi Yokota[3], Kenji Oritani[4], Toshio Watanabe[5], Yuzuru Kanakura[3] & Itaru Matsumura [1]

Mutated receptor tyrosine kinases (MT-RTKs) such as internal tandem duplication of FMS-like tyrosine kinase 3 (FLT3 ITD) and a point mutation KIT D816V are driver mutations for acute myeloid leukemia (AML). Clathrin assembly lymphoid myeloid leukemia protein (CALM) regulates intracellular transport of RTKs, however, the precise role for MT-RTKs remains elusive. We here show that CALM knock down leads to severely impaired FLT3 ITD- or KIT D814V-dependent cell growth compared to marginal influence on wild-type FLT3- or KIT-mediated cell growth. An antipsychotic drug chlorpromazine (CPZ) suppresses the growth of primary AML samples, and human CD34$^+$CD38$^-$ AML cells including AML initiating cells with MT-RTKs *in vitro* and *in vivo*. Mechanistically, CPZ reduces CALM protein at post transcriptional level and perturbs the intracellular localization of MT-RTKs, thereby blocking their signaling. Our study presents a therapeutic strategy for AML with MT-RTKs by altering the intracellular localization of MT-RTKs using CPZ.

[1] Department of Hematology and Rheumatology, Kindai University Faculty of Medicine, Osaka-sayama, Osaka, Japan. [2] Division of Hematological Malignancy, National Cancer Center Research Institute, Chuo, Tokyo, Japan. [3] Department of Hematology and Oncology, Osaka University Graduate School of Medicine, Suita, Osaka, Japan. [4] Department of Hematology, International University of Health and Welfare, Narita, Chiba, Japan. [5] Department of Biological Science, Graduate School of Humanities and Sciences, Nara Women's University, Nara, Nara, Japan. [6]These authors contributed equally: Shinya Rai, Hirokazu Tanaka. ✉email: htanaka@med.kindai.ac.jp

Activating mutations in receptor tyrosine kinases (RTKs) have been detected in various types of malignant diseases such as lung cancer, hepatocellular carcinoma, and acute myeloid leukemia (AML)[1]. These mutated RTKs (MT-RTKs) trigger the aberrant activation of signal pathways involved in cancer development and progression. Upon ligand binding, wild-type (WT) RTKs are internalized and transferred to early endosomes with clathrin-coated vesicles (CCVs) as a cargo, where they transmit proper signals to downstream molecules. Then, a part of WT-RTKs is sorted back to the plasma membrane (PM) via recycling endosomes, while the remaining part is transported to late endosomes and consequently degraded at lysosomes. However, the intracellular trafficking of MT-RTKs is spatially and functionally distinct from that of the WT-RTKs[2–5]. Although MT-RTKs are also carried by CCVs and follow the same route to early endosomes as was the case with WT-RTKs, most of MT-RTKs remain at endoplasmic reticulum (ER) or endolysosomes, where they transmit oncogenic signals persistently and cause malignant diseases[6–8]. In addition, degradation of MT-RTKs is retarded and their recycling is enhanced[2–5].

Fms-like tyrosine kinase 3 (FLT3) also known as CD135 is a receptor for FLT3 ligand (FL), which is expressed on hematopoietic stem/progenitor cells (HSC/HPCs) and plays crucial roles in their growth and survival[9–11]. Internal tandem duplication of FLT3 (FLT3 ITD) is detectable in one-third of AML patients and point mutations in the tyrosine kinase domain (TKD) in about 10% of AML patients, both of which are considered to be causative mutations of AML. In addition, FLT3 ITD has been shown to be a poor prognostic factor for AML, while the significance of TKD remains controversial. FLT3 ITD is localized at ER, where it aberrantly activates STAT5, upregulates Pim-1/2 expression, and evades negative regulation[6,12].

KIT, also known as CD117, is a receptor for stem cell factor (SCF), which is expressed on the surface of HSC/HPCs and several other cell types including primordial germ cells and interstitial cells of Cajal[13,14]. Upon SCF ligation, KIT expressed on HSC/HPCs transmits intracellular signals requisite for their survival and proliferation. A point mutation in the KIT gene at amino acid 816 (KIT D816V) is an activating mutation that is found in about 30% of patients in core binding factor (CBF)-AML. Although AML with t(8;21)(q22;q22) is classified into a good prognostic group, the presence of KIT D816V has been shown to be a poor prognostic factor[15].

CALM (clathrin assembly lymphoid myeloid leukemia) encoded by the PICALM gene was first identified as a component of the fusion gene CALM/AF10 resulting from the chromosomal translocation t(10;11) (p13;q14) in AML cells. This fusion gene is also found in acute lymphoblastic leukemia (ALL) and malignant lymphomas[16]. Several studies showed that CALM/AF10 reveals oncogenic activities primarily through AF10 but not through CALM[17,18].

CALM regulates the size and maturation of CCVs by recognizing membrane curvature[19]. ANTH domain of CALM at N-terminus plays an important role in the direct recognition of cargo proteins[20]. We previously reported that CALM deficient ($CALM^{-/-}$) mice exhibit retarded growth in utero and suffer from severe anemia due to the impaired clathrin-mediated endocytosis of transferrin in immature erythroblast[21]. In addition, severely reduced number of early HPCs coupled with numerous morphologic and functional defects are observed in the peripheral blood (PB) of $CALM^{-/-}$ mice[21]. These findings are largely consistent with the phenotypes of fit1 mice harboring a nonsense point mutation in the PICALM gene[22,23]. Furthermore, we showed that CALM is essential for CCV formation and plays an important role in the intracellular trafficking of KIT from early to late endosomes in hematopoietic cells[24]. In this study, we also found that KIT-mediated cellular growth was partially impaired in $CALM^{-/-}$ HSC/HPCs. These results suggest that intracellular trafficking might be a therapeutic target in AML cells with MT-RTKs.

We here report that a widely used antipsychotic drug, chlorpromazine (CPZ), which is known to inhibit CCV formation[25], suppresses CALM protein levels and alters the intracellular localization of MT-RTKs, thereby inhibiting the growth of AML cells with MT-RTKs in vitro and in vivo. These results propose a possibility that altering the intracellular localization of MT-RTKs using CPZ or more potent analogues will be a therapeutic strategy for AML with MT-RTKs.

## Results

**CALM is required for oncogenic signals in MT-RTK AML cells.** To examine the role of CALM in RTKs-dependent growth and survival of hematopoietic cells, we utilized a murine pro-B cell line, Ba/F3, of which growth and survival are absolutely dependent on IL-3. For this purpose, we stably introduced FLT3 (FLT3 WT or FLT3 ITD) and KIT (KIT WT or KIT D814V that corresponds to human D816V in the murine gene) into Ba/F3 cells and generated Ba/F3 sublines as follows: Ba/F3-FLT3 WT, Ba/F3-FLT3 ITD, Ba/F3-KIT WT, and Ba/F3-KIT D814V.

Next, we stably knocked down (KD) CALM in these clones by delivering retrovirus particles carrying shRNA specific to CALM (CALM shRNA) or scrambled (SCR) shRNA as a control. Western blotting showed that CALM shRNA reduced CALM protein levels to <25% compared with SCR shRNA (Supplementary Fig. 1a). Both Ba/F3-FLT3 WT/SCR and Ba/F3-KIT WT/SCR cells proliferated in response to their cognate ligands, FL and SCF, respectively. On the other hand, Ba/F3-FLT3 ITD/SCR and Ba/F3-KIT D814V/SCR cells proliferated under IL-3-, FL-, and SCF-deprived conditions (Fig. 1a). Importantly, CALM KD did not influence the IL-3-dependent growth of Ba/F3-FLT3 WT and Ba/F3-KIT WT, however, the FL-dependent growth of Ba/F3-FLT3 WT and SCF-dependent growth of Ba/F3-KIT WT were slightly reduced by CALM KD (Fig. 1a). In this condition, in accord with our previous report[24], FL-induced phosphorylation of FLT3 WT, SCF-induced phosphorylation of KIT WT, and phosphorylation of their downstream molecules (STAT5, ERK, and Akt) were not impaired, while phosphorylation of Akt was slightly augmented and prolonged in Ba/F3-FLT3 WT and Ba/F3-KIT WT by CALM KD (Supplementary Fig. 1b).

In contrast, MT RTKs-dependent growth was considerably suppressed by CALM KD in Ba/F3-FLT3 ITD and Ba/F3-KIT D814V cells under cytokine-deprived conditions (Fig. 1a). As for this mechanism, autophosphorylation of FLT3 ITD and KIT D814V and phosphorylation of their downstream molecules (STAT5 for FLT3 and ERK1/2 and Akt for KIT) were suppressed by CALM KD in Ba/F3-FLT3 ITD and Ba/F3-KIT D814V cells (Supplementary Fig. 1c).

Consistent with these in vitro findings, tumorigenic activities of Ba/F3-FLT3 ITD and Ba/F3-KIT D814V were severely suppressed by CALM KD in transplanted mice (Supplementary Fig. 1d–f), resulting in their prolonged survival (Supplementary Fig. 1g, h).

These results indicate that CALM plays a crucial role in MT-RTKs-dependent growth but not in WT-RTKs-dependent growth, and suggest that ligand-activated WT-RTKs and MT-RTKs are differently regulated by CALM.

To examine whether these findings are applicable to AML cells, we KD CALM in an inducible manner (CLAM iKD) in MV4-11 (with FLT3 ITD), HMC-1 (with KIT D816V) and HL-60 (with WT-FLT3 and KIT) cells with a doxycycline (DOX) inducible system, in which CALM shRNA expression was induced by the

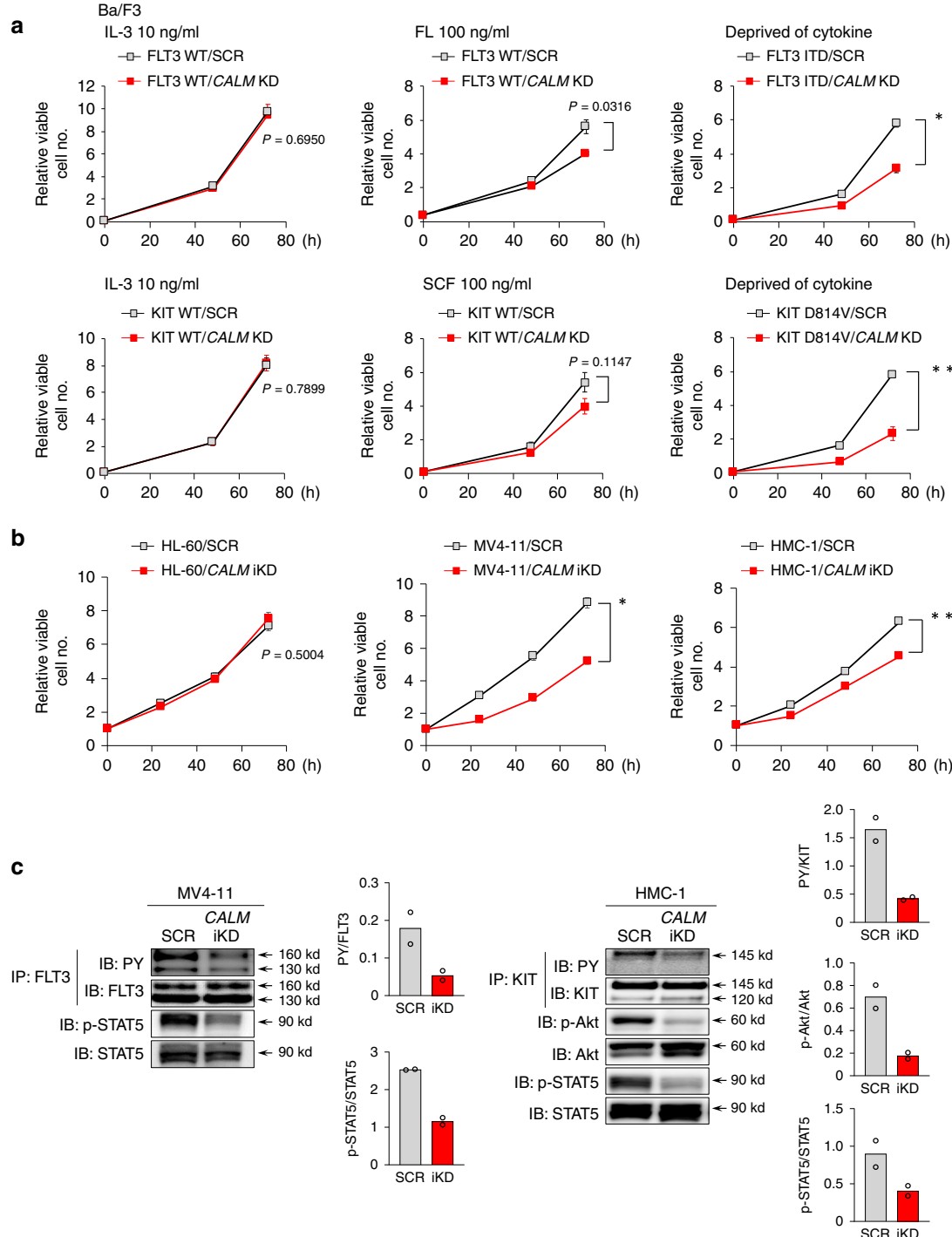

**Fig. 1 *CALM* shRNA severely impairs the growth of hematopoietic cells with MT-RTKs. a** CALM was knocked down in Ba/F3-FLT3 WT, Ba/F3-FLT3 ITD, Ba/F3-KIT WT, and Ba/F3-KIT D814V by shRNA specific to *CALM* (*CALM* shRNA) or scrambled (SCR) shRNA as a control. These clones were cultured under various conditions to assess the influences of *CALM* KD on IL-3-dependent growth (left panel), FL- and SCF-dependent growth (center panel), and FLT3 ITD- and KIT D814V-dependent growth (right panel). The growth of these cells was assessed at the indicated points. Data shown are the mean ± SEM from three independent experiments. Two-sided unpaired Student's *t* test, *$*p = 0.0018$, **$**p = 0.0018$. **b** *CALM* was knocked down with SCR shRNA as a control in AML cell lines, MV4-11, HMC-1, and HL-60, with a doxycycline (DOX) inducible system. The growth of these cells was monitored until 72 h after the start of DOX treatment. Data shown are the mean ± SEM from three independent experiments. Two-sided unpaired Student's *t* test, *$*p = 0.0013$, **$**p = 0.0003$. **c** Tyrosine-phosphorylated status of FLT3, KIT and their downstream STAT5, and Akt were investigated by immunoprecipitation and/or immunoblot analyses using the indicated cells. Representative images of two independent experiments are shown. Densitometry analysis was carried out by Image Quant TL, and data is from two independent experiments.

addition of DOX into the culture medium. Western blotting showed that induction of *CALM* shRNA by DOX resulted in more than 80% reduction of CALM protein in MV4-11, HMC-1 and HL-60 cells (Supplementary Fig. 1a). Compared with SCR shRNA, *CALM* iKD suppressed the growth of MV4-11 cells by about 40% and that of HMC-1 by about 30%, while it hardly influenced the growth of HL-60 cells (Fig.1b). In addition, phosphorylation of FLT3 and STAT5 was severely suppressed by *CALM* iKD in MV4-11 cells (Fig. 1c). Similarly, *CALM* iKD suppressed phosphorylation of KIT, Akt and, STAT5 in HMC-1 cells. These results indicate that CALM is required for oncogenic signals in AML cells with MT-RTKs.

**CPZ impairs MT-RTKs-dependent growth of AML cells** in vitro. We treated leukemia cells with CPZ, an antipsychotic drug that has been widely used for more than 50 years. CPZ has various pharmacological activities. Among them, CPZ induces clathrin misassemble on internal structures, thereby inhibiting CCV formation[25]. As shown in Fig. 2a, CPZ dose-dependently suppressed the growth of Ba/F3-FLT3 ITD with IC50 6.940 μM and that of Ba/F3-KIT D814V with IC50 6.942 μM. To characterize the mechanism of CPZ-induced cell death, we performed annexin V staining of Ba/F3-FLT3 ITD cells after culture with or without CPZ. As shown in Supplementary Fig. 2, CPZ treatment increased apoptotic cells detected as an Annexin V-positive fraction.

However, CPZ didn't further augment (no additive effect) growth inhibitory effects of *CALM* KD in Ba/F3-FLT3 ITD and Ba/F3-KIT D814V cells (Fig. 2b), suggesting that CPZ suppresses MT-RTKs-dependent growth of Ba/F3 cells through the inhibition of CALM.

We also evaluated antileukemic activities of CPZ using several human myeloid cell lines, including FLT3 ITD[+] MV4-11, KIT D816V[+] HMC-1, and other cell lines with FLT WT/KIT WT (BALL-1, HL-60, TALL-1, KG-1, THP-1, Kasumi, and K562). When these cell lines were treated with 7.0 μM CPZ for 72 h, CPZ drastically (more than 90%) inhibited the growth and survival of MV4-11 and HMC-1 but showed only marginal (0–40%) inhibitory effects on leukemia cell lines with FLT3 WT/KIT WT (Fig. 2c).

We next examined the effects of CPZ on the growth of primary AML cells with FLT3 WT/KIT WT (*n* = 5), FLT3 ITD (*n* = 3), and KIT D816V (*n* = 1). These cells were cultured in the presence of SCF, FL, and TPO with or without 7.0 μM CPZ for 72 h. As expected, CPZ inhibited the growth of AML cells harboring either FLT3 ITD (#1, #5, and #7) or KIT D816V (#10), while it showed no effect (#11 and #16) or limited effects (#12, #17, and #19) on AML cells with FLT3 WT/KIT WT (Fig. 2d).

We also examined the effects of CPZ on human CD34[+]CD38[−] (hCD34[+]hCD38[−]) AML cells with FLT3 ITD. In our xeno-transplantation model using NOD/Scid/IL2Rγ-KO (NOG) mice, when this fraction was isolated from the mice that developed AML after the 1st transplantation of primary AML cells (Cases #2, #3, #5 and #8) and transplanted into the 2nd mice, all secondary transplanted mice developed AML (Supplementary Fig. 3), indicating that this fraction contains AML initiating cells. This fraction (from Cases #3, #5) was also sensitive to CPZ, suggesting a possibility that CPZ may have a potential to eradicate AML initiating cells (Fig. 2e).

**CPZ targets MT-RTK AML cells through CALM protein depletion**. CPZ also has been shown to act as an antagonist of dopamine receptor (DR), serotine receptor (5-HTR), and histamine receptor (HR), and DR and 5-HTR have been reported as therapeutic targets for leukemic stem cells[26,27]. Thus, we tested

whether anti-AML activities of CPZ were mediated by the interference of CCV formation or by inhibition of DR and/or 5-HTR. For this purpose, we treated AML cell lines with WT-RTKs (HL-60, Kasumi, THP-1, KG-1 and K562) or with MT-RTKs (MV4-11 and HMC-1) with a DR antagonist, Thioridazine (THIO) and a 5-HTR antagonist, Methiothepin (METHIO), respectively. As shown in Fig. 3a, all tested AML cell lines showed some sensitivity to both THIO and METHIO at various degrees. However, MV4-11 and HMC-1 were rather resistant to both agents (growth/survival inhibition by THIO 10% and 27%, respectively; by METHIO 32% and 28%, respectively) compared with HL-60, Kasumi, THP-1, KG-1, and K562. Consistent with the reported elsewhere[27], when we evaluated apoptotic cells as an Annexin V-positive fraction, HL-60 cells that express high levels of DR and 5-HTR were highly sensitive to THIO and METHIO but not to CPZ compared with MV4-11 and HMC-1 cells (Fig. 3b).

Importantly, CPZ treatment dose-dependently reduced CALM protein levels in MV4-11 and HMC-1 cells (Fig. 3c), whereas it increased *CALM* mRNA levels in HMC-1 cells (Fig. 3d), suggesting that CPZ reduced CALM protein at posttranscriptional levels. In contrast, CPZ treatment increased clathrin heavy chain protein levels and didn't influence HSPA8 protein levels in both cell lines (Fig. 3c).

We further examined the effects of CPZ on the growth of MV4-11 and HMC-1 cells, in which CALM was KD. In contrast to the observed in *CALM* KD Ba/F3-FLT3 ITD and Ba/F3-KIT D814V cells (Fig. 2b), CPZ further suppressed the growth of *CALM* iKD MV4-11 and HMC-1 cells (30% by *CALM* iKD and 55% by *CALM* iKD + CPZ in HMC-1; 40% by *CALM* iKD and 60% by *CALM* iKD+CPZ in MV4-11) (Fig. 3e).

Together, these results suggest that, although CPZ selectively targets AML cells with MT-RTKs mainly through CALM protein depletion, inhibition of other molecules such as DR and 5-HTR might, to some extent, contribute to full anti-AML activities of CPZ depending on cellular contexts. Of note, because CPZ reduced CALM protein levels in MV4-11 and HMC-1 cells (Fig. 3c), CPZ might further deplete residual CALM protein escaped from iKD in these cells.

**CALM depletion by CPZ perturbs the localization of MT-RTKs**. We investigated the interaction between CALM and FLT3 ITD in MV4-11 cells by immunofluorescence microscopy. At first, we examined the intracellular distribution of CALM using various compartment markers such as protein disulfide isomerase (PDI) (for ER), Golgi marker protein 130 (GM130) (for cis-Golgi), early endosome antigen-1 (EEA1) (for endosome), lysosome-associated membrane proteins 1 (LAMP1) (for endo-lysosome), and Rab11 (for recycling endosome). As a result, CALM was most abundantly co-localized with PDI (33.5% of total amount) and substantially with LAMP1 (20.8%) and EEA1 (15.3%) and partly with Rab11 (5.1%) and GM130 (3.1%) in MV4-11 cells (supplementary Fig. 4a). When *CALM* iKD MV4-11 cells were treated with DOX, the amount of CALM was effectively reduced by iKD (Fig. 4a) as previously shown in Supplementary Fig. 1a. Also, CPZ treatment severely reduced the amount of CALM (Fig. 4a), of which finding was largely consistent with the result shown in Fig. 3c. Without any treatment, nearly 80% of FLT3 ITD was co-localized with CALM (Fig. 4a). However, the proportion of FLT3 ITD co-localized with CALM was severely reduced from 79.1 to 4.2% by *CALM* iKD and to 6.9% by CPZ treatment due to CALM depletion (Fig. 4a, c).

In untreated cells, FLT3 was co-localized with PDI (30.1%), with Rab11 (21.9%), and somewhat with GM130 (4.9%), EEA1 (8.6%), and LAMP1 (7.8%) (Fig. 4b, c, Supplementary Fig. 4b),

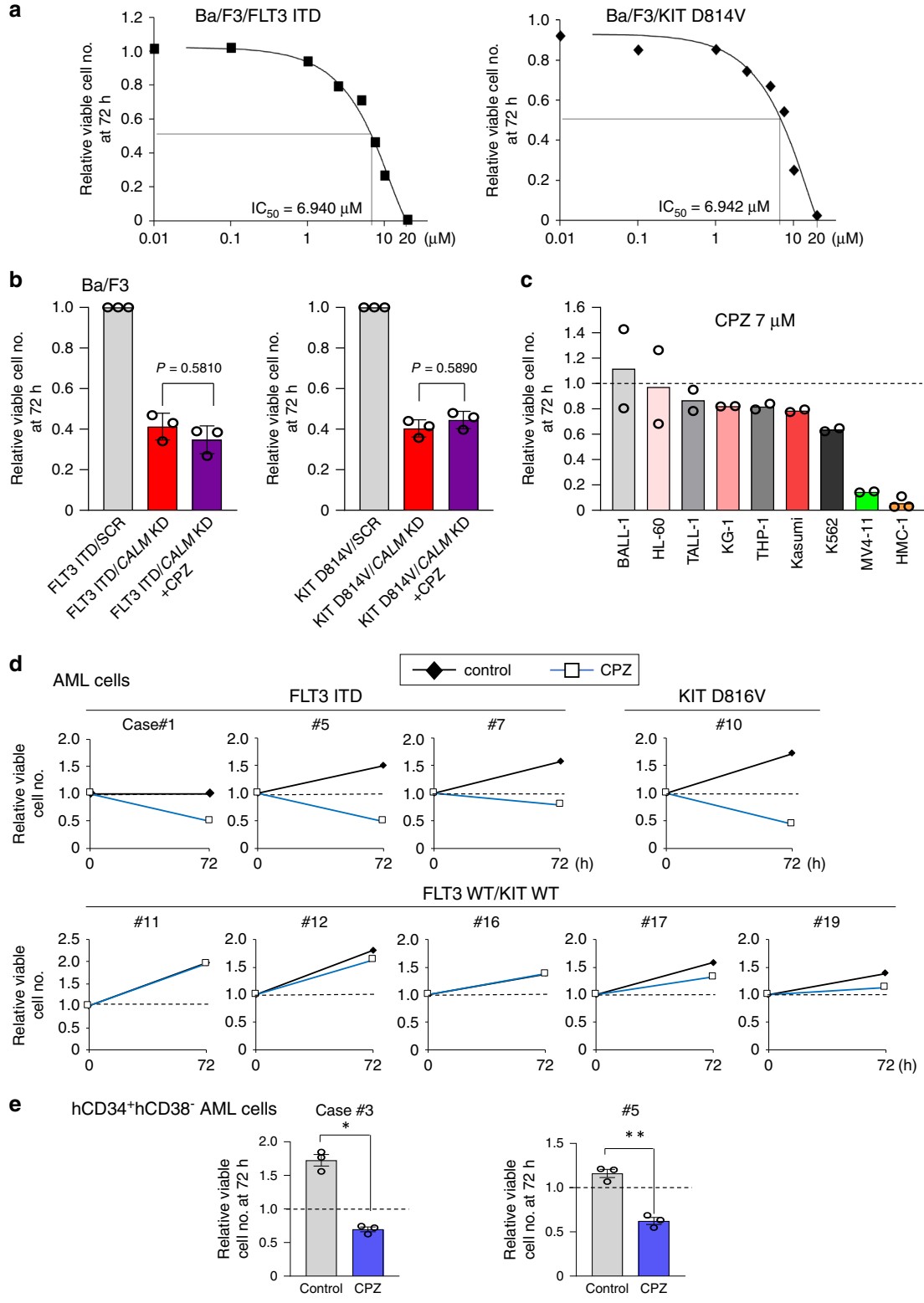

suggesting that it is preferentially present at ER. However, *CALM* iKD or CPZ treatment reduced the co-localization of FLT3 ITD with PDI (to 7.1% by *CALM* iKD; to 6.1% by CPZ) (Fig. 4b, c). In contrast, *CALM* iKD or CPZ treatment neither reduced or augmented the co-localization of FLT3 ITD with Rab11 (to 24.7% by *CALM* iKD; to 23.7% by CPZ), GM130 (to 4% by *CALM* iKD; to 4.2% by CPZ), EEA1 (to 6.7% by *CALM* iKD; to 7.4% by CPZ), or LAMP1 (to 6.6% by *CALM* iKD; to 6.6% by CPZ) (Fig. 4c,

Supplementary Fig. 4b). Consistent with this observation, the amount of FLT3 ITD was severely reduced by *CALM* iKD or CPZ treatment in the isolated PDI-positive ER fraction compared with untreated cells (Fig. 5a). Moreover, although the total amount of FLT3 ITD in the whole cellular lysates was not influenced by CPZ treatment, phosphorylation of FLT3 ITD and STAT5 was inhibited by CPZ in a dose-dependent manner (Fig. 5b) as observed in *CALM* iKD MV4-11 cells (Fig. 1c). These results are

**Fig. 2 CPZ selectively inhibits the growth/survival of AML cells with MT-RTKs. a** Ba/F3-FLT ITD and Ba/F3- KIT D814V cells were cultured with various concentrations of CPZ under cytokine-deprived condition. Cell growth was evaluated with the Cell Titer Glo Reagent. IC50s of CPZ for these clones were determined as described in material and methods. **b** The left panel shows the relative growth rate of Ba/F3-FLT3 ITD/*CALM* KD cultured with or without 7.0 µM CPZ using Ba/F3-FLT3 ITD/SCR as a reference. The right panel shows the relative growth rate of Ba/F3-KIT D814V/*CALM* KD cultured with or without 7.0 µM CPZ using Ba/F3-KIT D814V/SCR as a reference. The results indicate the mean ± SEM from three independent experiments. One-way ANOVA multiple comparisons with Bonferroni correction, $p = 0.5810$ (left panel), $p = 0.5890$ (right panel). **c** Various leukemia cell lines were cultured in the presence or absence of 7.0 µM CPZ for 72 h. Relative growth rates of CPZ-treated cells were calculated with respective CPZ-untreated cells as a reference. Data are from two independent experiments for BALL-1, HL-60, TALL-1, KG-1, THP-1, Kasumi, K562, and MV4-11 and three independent experiments for HMC-1. **d** BMMNCs containing >70% AML cells were isolated from nine patients with the indicated status of *FLT3* and *KIT* genes. These cells were cultured in the presence of SCF, FL, and TPO with or withou7.0 µM of CPZ for 72 h, and viable cell numbers were determined. Relative viable cell numbers of CPZ-treated cells were calculated using respective control cells as a reference. **e** Human CD34$^+$CD38$^-$ (hCD34$^+$hCD38$^-$) AML cells were sorted from BMMNCs of two patients by flow cytometry. These cells were cultured in the presence of SCF, FL, and TPO with or without 7.0 µM CPZ for 72 h, and viable cell numbers were quantified. The results indicate the mean ± SEM from three independent experiments. Two-sided unpaired Student's $t$ test, $*p = 0.0004$, $**p = 0.0009$.

in agreement with a previous report indicating that localization at ER is essential for FLT3 ITD to activate STAT5[12].

In untreated HMC-1 cells, CALM was co-localized with LAMP1 (20.8%), EEA1 (15.2%), PDI (12.9%), GM130 (4.3%), and Rab11 (1.9%) (Supplementary Fig. 4c). As was the case with MV4-11 cells, CALM iKD or CPZ treatment apparently reduced CALM protein (Fig. 4d). Although 86.5% of KIT D816V was co-localized with CALM, this co-localization was impaired by *CALM* iKD or CPZ treatment due to CALM depletion (to 7.1% by CALM iKD; to 12.2% by CPZ) (Fig. 4d, f).

We next investigated the intracellular localization of KIT D816V in HMC-1 cells. Without any treatment, KIT D816V was co-localized with LAMP1 (26.4%) (Fig. 4e, f), with EEA1 (17%), and somewhat with PDI (13.7%), GM130 (10.6%), and Rab11 (4.8%), suggesting that it is enriched in endolysosome. In contrast, *CALM* iKD by DOX treatment or CPZ treatment obviously reduced co-localization of KIT D816V with LAMP1 (to 7.2% by *CALM* iKD; to 8% by CPZ), with EEA1 (to 6.3% by *CALM* iKD; to 6.1% by CPZ), and with PDI (to 8.4% by *CALM* iKD; to 8.3% by CPZ), GM130 (to 7.1% by *CALM* iKD; to 5.9% by CPZ) except for the co-localization with Rab11 (to 4.6% by *CALM* iKD; to 4.3% by CPZ) (Fig. 4f, Supplementary Fig. 4d). In accord with these findings, the amount of KIT D816V in the endolysosome fraction coimmunoprecipitated with LAMP1 was significantly reduced by *CALM* iKD or CPZ treatment compared with untreated HMC-1 cells (Fig. 5c). However, immunoblot analysis using whole cellular lysates showed that the total amount of KIT was not reduced but rather increased by CPZ treatment. We also found that phosphorylation of KIT D816V and Akt was inhibited by CPZ in a dose-dependent manner (Fig. 5d). This result is largely consistent with a previous paper showing that localization at endolysosomes is critical for KIT D816V to activate Akt[8].

Together, these results indicate that CPZ reduces CALM protein and perturbs the intracellular localization of FLT ITD and KIT D816V, thereby inhibiting their oncogenic signals.

**CPZ inhibits leukemic activities of MT-RTK AML cells** in vivo. We tested antileukemic activities of CPZ in a xenograft mouse model using primary AML cells (case #2). We transplanted $1 \times 10^6$ BM cells (including more than 70% AML cells) with FLT3 ITD into sub-lethally irradiated immunodeficient NOG mice. Daily intraperitoneal injection of CPZ (at 10 mg/kg achieving the serum concentration relevant to the in vitro experiments) was started from 10 days after transplantation and continued for 8 weeks. Pharmacokinetic analysis showed that the plasma concentrations of CPZ by this dose were almost equivalent to those achieved in the plasma of patients receiving CPZ as an antipsychotic drug (Supplementary Fig. 5). Nine weeks after

transplantation, mice were euthanized and subjected to analyses. Although femurs from control mice were macroscopically pale due to severely impaired hematopoiesis, those from CPZ-treated mice exhibited a normal color indicative of recovery of hematopoiesis (Fig. 6a). The mean percentage of hCD45$^+$ (%hCD45$^+$) cells was 83.8% in the BM of control mice, which was reduced to 6.03% by CPZ treatment. Also, CPZ treatment reduced the mean % of hCD33$^+$hCD56$^+$ AML cells from 72.7 to 3.83% (Fig. 6b).

We analyzed the in vivo effect of CPZ on primary AML cells from 19 patients (Age 17–81, median 48, male/female 12/7), including AML cells with FLT3 WT/KIT WT (ten cases), FLT3 ITD (eight cases), and KIT D816V (one case) (Supplementary Table 1). In control mice, mean %hCD45$^+$ cells ranged from 0.56% to nearly 100% at 8–10 weeks after transplantation, indicating a large interindividual variability in the leukemogenic potential of AML samples. As seen in case #2, CPZ treatment effectively inhibited the growth of AML samples with MT-RTKs and statistically significant differences were observed in 6/8 cases with FLT3 ITD and 1/1 case with KIT D816V. In contrast, although CPZ exhibited some antileukemic activities on AML cells with FLT3 WT/KIT WT (Fig. 6c), a statistically significant effect was observed in only one case (case #17) out of ten analyzed samples.

Histopathological analysis revealed that, whereas femurs from control mice transplanted with FLT3 ITD$^+$ AML cells (from case #2) were packed with hCD45$^+$ cells with numerous hCD34$^+$ cells, those from CPZ-treated mice showed recovery of normal hematopoiesis with a small number of hCD45$^+$ cells and rare hCD34$^+$ cells. Importantly, while a large number of hCD34$^+$ cells were detected in the endosteal BM niche in control mice, CPZ treatment eliminated these cells almost completely (Fig. 6d). This observation suggests a possibility that CPZ may eradicate FLT ITD$^+$ AML initiating cells in vivo. We also conducted RT-PCR analysis to characterize engrafted BM cells. As shown in Fig. 6e, both FLT3 WT and ITD fragments were amplified from the pre-transplant AML sample. Similarly, both fragments were amplified from the BM of control mice after transplantation. In contrast, only FLT3 WT (but not FLT3 ITD) was amplified from the BM of CPZ-treated mice, indicating that the majority of hCD45$^+$ cells observed in the BM of CPZ-treated mice derived from engrafted normal human HSCs.

To further evaluate the therapeutic potential of CPZ on established AML, we retarded CPZ administration until transplanted mice developed leukemia. In these experiments, we transplanted $1 \times 10^6$ BM cells from case #3 (FLT3 ITD$^+$) into irradiated NOG mice. Six weeks after transplantation, the mean % hCD45$^+$ cells in the BM cells was 19.4% (Supplementary Fig. 6a), indicating that transplanted AML cells were engrafted and caused leukemia in the recipient mice. Then, we treated these mice with

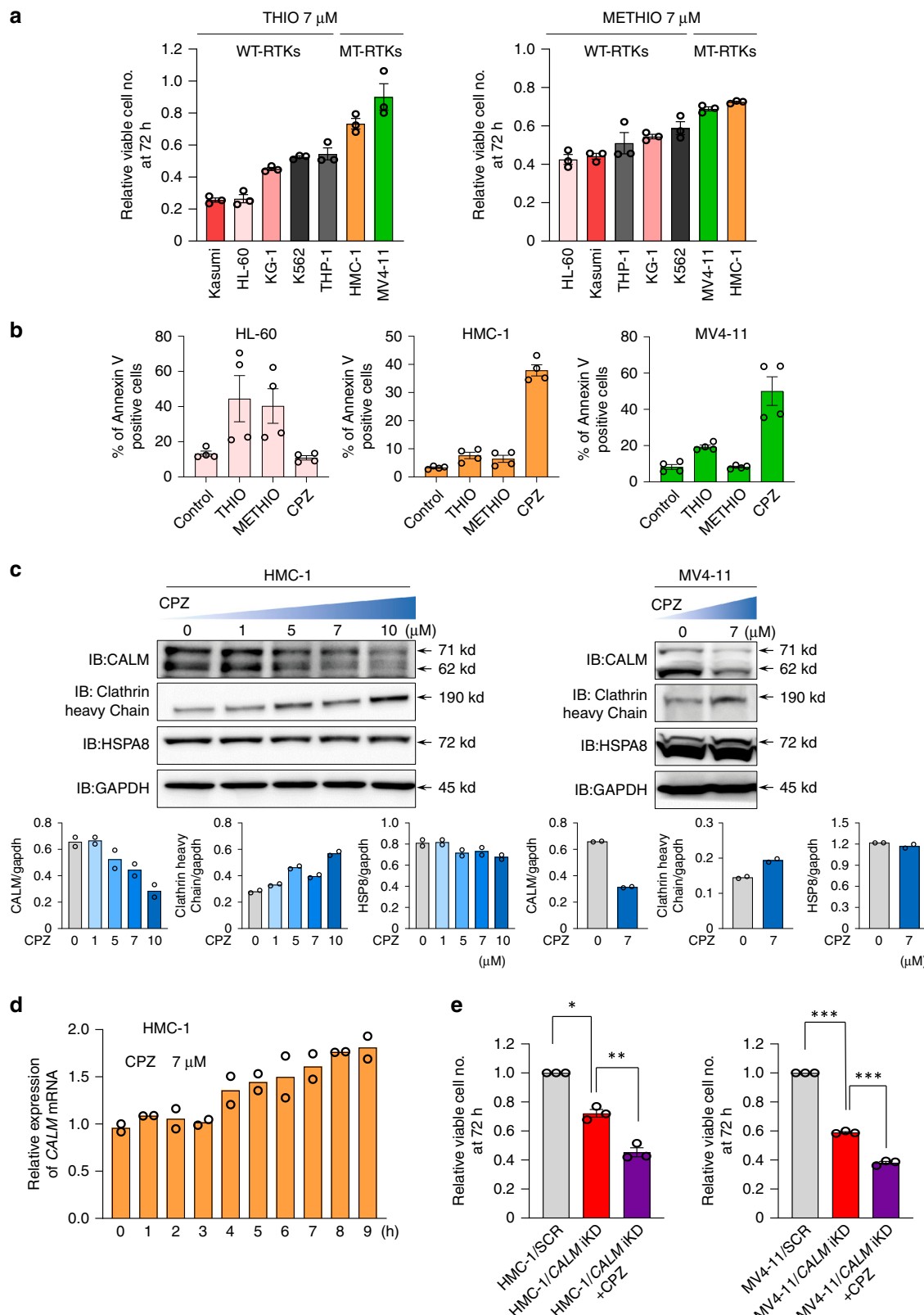

CPZ or normal saline (NS) (as a control) for 4 weeks. All of the control mice died of leukemia 10 weeks after transplantation. In contrast, all of the CPZ-treated mice survived until 10 weeks after transplantation. Also, CPZ treatment effectively reduced the mean %hCD45$^+$cells in BM from 14.5 to 1.5% ($n = 4$), which was accompanied by the restoration of normal murine hematopoiesis

(% of murine CD45$^+$ cells: from 18.6 to 58.6%) (Supplementary Fig. 6b, c).

To further assess the toxic effects of CPZ on normal hematopoiesis, we transplanted $1 \times 10^4$ normal human cord blood (CB) CD34$^+$ cells into NOG mice and treated these mice with CPZ or NS (as a control) from 3 days after transplantation

**Fig. 3 CPZ selectively targets AML cells with MT-RTKs mainly through CALM inhibition. a** Various leukemia cell lines were cultured in the presence or absence of 7.0 μM serotonin receptor antagonist (METHIO) or dopamine receptor antagonist (THIO) for 72 h and viable cell numbers were evaluated. Relative growth ratio was calculated using respective untreated cells as a reference. Data indicates the mean ± SEM from three independent experiments. **b** After 72 h treatment with THIO, METHIO, or CPZ, apoptotic cells were detected as Annexin V-positive cells. The results obtained from the indicated cells are shown as the mean ± SEM from four independent experiments. **c** HMC-1 and MV4-11 cells were cultured with or without CPZ at the indicated concentrations for 24 h. Immunoblot analyses were performed with the indicated Abs. Densitometry analysis was carried out by Image Quant TL, and data are from two independent experiments. **d** HMC-1 cells were cultured with 7.0 μM CPZ for the indicated times, and *CALM* mRNA levels were quantified by qRT-PCR. Figure shows the mean from two independent experiments. **e** The relative growth rates of HMC-1/*CALM* iKD (left panel) and MV4-11/*CALM* iKD (right panel) cultured with or without 7.0 μM CPZ were evaluated with HMC-1/SCR and MV4-11/SCR as references, respectively. The results indicate the mean ± SEM from three independent experiments. One-way ANOVA multiple comparisons with Bonferroni correction, $*p = 0.0005$, $**p = 0.0006$, $***p < 0.0001$.

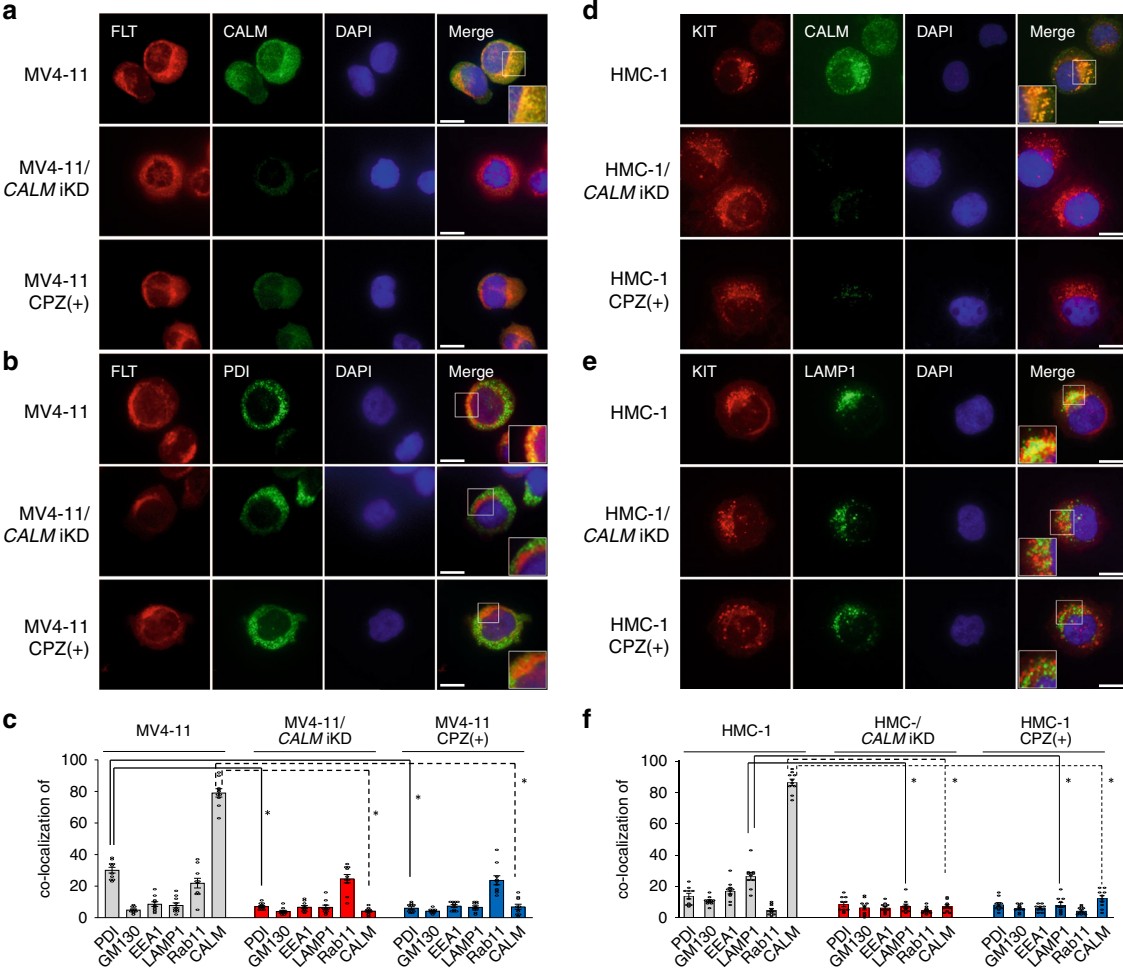

**Fig. 4 CALM depletion perturbs the intracellular localization of MT-RTKs. a** Parental, *CALM* iKD, and 18 h CPZ-treated MV4-11 cells were subjected to immunofluorescence microscopy. The cells were co-stained with the anti-FLT3 Ab (red), anti-CALM Ab (green), and 4′,6-diamidino-2-phenylindole (DAPI) (blue). Scale bars, 20 μm. The right lower panels show regions with higher magnification. Images are representative of three independent experiments. **b** Parental, *CALM* iKD, and 18 h CPZ-treated MV4-11 cells were co-stained with the anti-FLT3 Ab (red), anti-protein disulfide isomerase (PDI) Ab (green), and DAPI (blue). Scale bars, 20 μm. The right lower panels show regions with higher magnification. Images are representative of three independent experiments. **c** The graph bars show the percentages of co-localization of FLT3 with PDI, Golgi matrix protein 130 (GM130), early endosome antigen-1 (EEA1), lysosome-associated membrane protein-1 (LAMP1), Rab11 and CALM. Results (%) represent the means ± SEM from ten cells. One-way ANOVA multiple comparisons with Bonferroni correction, $*p < 0.0001$. **d** Parental, *CALM* iKD, and 24 h CPZ-treated HMC-1 cells were subjected to immunofluorescence microscopy. The cells were co-stained with the anti-KIT Ab (red), anti-CALM (green) Ab, and DAPI (blue). Scale bars, 20 μm. Regions with higher magnification are shown in left lower panels. Images are representative of three independent experiments. **e** Parental, *CALM* iKD, and 24 h CPZ-treated HMC-1 cells were stained with the anti-KIT Ab (red), anti- LAMP1 Ab (green), and DAPI (blue). Scale bars, 20 μm. Regions with higher magnification are shown in left lower panels. Images are representative of three independent experiments. **f** The graph bars show the percentages of co-localization of KIT with PDI, GM130, EEA1, LAMP1, Rab11 and CALM with the means ± SEM from ten cells. One-way ANOVA multiple comparisons with Bonferroni correction, $*p < 0.0001$.

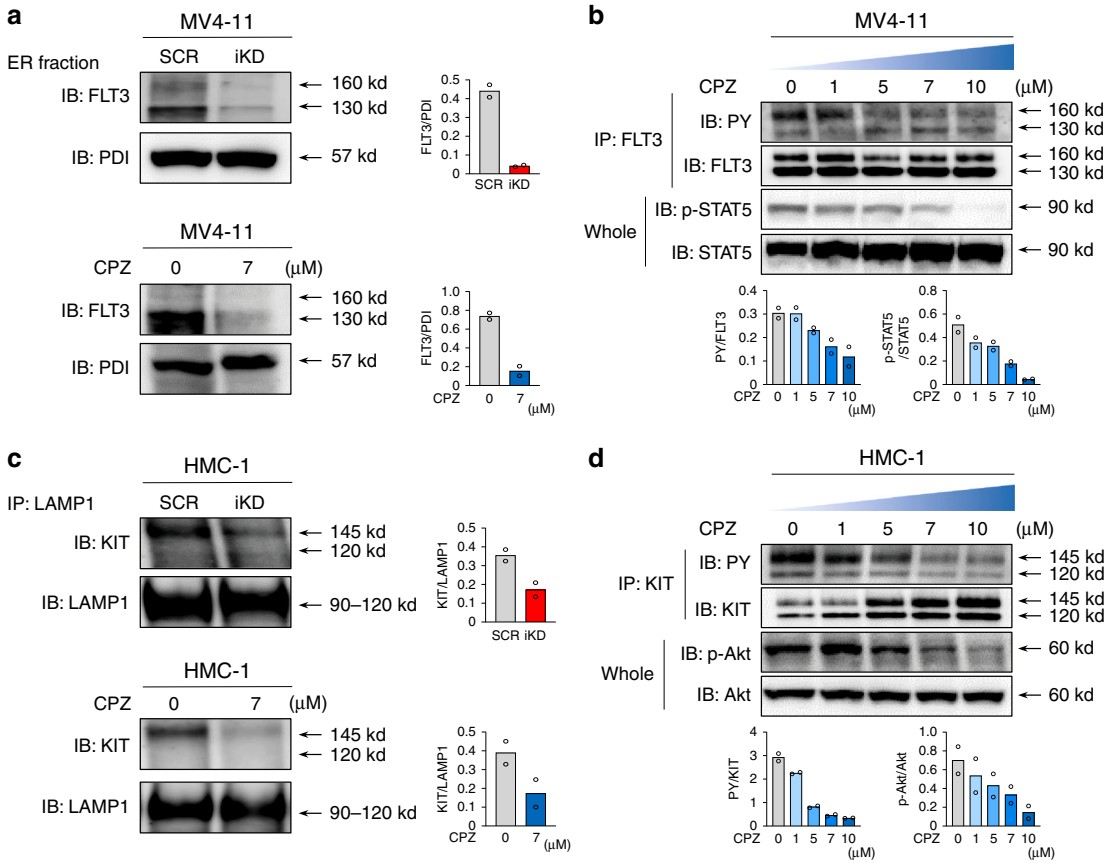

**Fig. 5 CPZ blocks compartment-dependent oncogenic signaling from MT-RTKs. a** PDI-positive ER fraction was isolated from *CALM* iKD, *CALM* SCR, CPZ-treated, and CPZ-untreated MV4-11 cells and subjected to immunoblot analyses. **b** MV4-11 cells were cultured with CPZ at the indicated concentrations for 18 h. Whole cell lysates or immunoprecipitated proteins were isolated and subjected to immunoblot analyses with the indicated Abs. **c** LAMP-1-positive protein was immunoprecipitation from the total lysates of *CALM* iKD, *CALM* SCR, CPZ-treated, or untreated HMC-1 cells, and subjected to immunoblot analyses using the indicated Abs. **d** HMC-1 cells were cultured for 24 h with or without CPZ at the indicated concentration, Phosphorylated status of KIT and Akt was assessed by immunoblot analyses on the whole lysates or immunoprecipitated proteins. These densitometry analyses were carried out using Image Quant TL, and data are from two independent experiments.

(Supplementary Fig. 7a). As a result, the mean %hCD45[+] cells was 26.1% in control mice and 29.3% in CPZ-treated mice 8 weeks after transplantation with no statistical difference (Supplementary Fig. 7b). In addition, similar percentages of hCD19[+] B cells and hCD33[+] myeloid cells were detected both in control and CPZ-treated mice (Supplementary Fig. 7c), indicating that CPZ doesn't affect the engraftment and repopulating capacity of normal HSCs.

## Discussion

In this study we found that CALM protein is critical for the growth and survival of leukemia cells harboring MT-RTKs but not for WT-RTKs-dependent growth. These results suggest that CALM and/or its associate proteins would somewhat differently regulate the trafficking of MT-RTKs and that of WT-RTKs. In accord with previous reports, we found that FLT3 ITD is predominantly localized at ER and KIT D816V at endolysosomes, where they activate themselves and their downstream molecules, STAT5 and Akt. However, when CALM was KD or these cells were treated with CPZ, these MT-RTKs became to be localized in the cytoplasm with a disorganized pattern, resulting in severely reduced phosphorylation of MT-RTKs themselves and of their downstream STAT5 and Akt. These results suggest that altering the intracellular localization of MT-RTKs will be a therapeutic strategy for AML with MT-RTKs. Although we tried to determine the precise localization of FLT3 ITD and KIT D816V after CALM

iKD or CPZ treatment, using several compartment markers such PDI (for ER), GM130 (for Golgi), EEA1 (for endosome), LAMP1 (for endolysosome), and Rab11 (for recycling endosome) in the immunofluorescence analyses, the co-localization pattern of these MT-RTKs didn't match those of any tested markers. Because CALM is a clathrin adapter, we speculated that perturbing its function by CALM iKD or CPZ could lead to accumulation of the mutated receptors on clathrin-positive structures, that originate at the PM or at late Golgi membranes. Furthermore, the reduced receptor localization at endolysosomes could result from inhibition of clathrin-mediated trafficking from the PM or the Golgi. However, we couldn't determine to which cellular compartments the mutated receptors are redistributed due to the limitation of the utilized markers. The presented images do not suggest PM accumulation, however GM130 is a cis-Golgi marker and does not represent all structures involved in Golgi-to-lysosome trafficking. So, further studies are needed to answer this issue. Because CPZ treatment severely reduced CALM protein in MV4-11 and HMC-1 cells, we assumed that the intracellular trafficking of MT-RTKs and their subsequent localization of MT-RTKs were totally impaired and disorganized by CPZ through downregulation of CALM protein.

CPZ has been used to treat psychotic disorders such as schizophrenia as an antagonist against dopaminergic receptors (D1–D4)[28]. CPZ also inhibits other receptors, including receptors for 5-HT, H1 histamine, α1 and α2 adrenaline, and M1 and M2

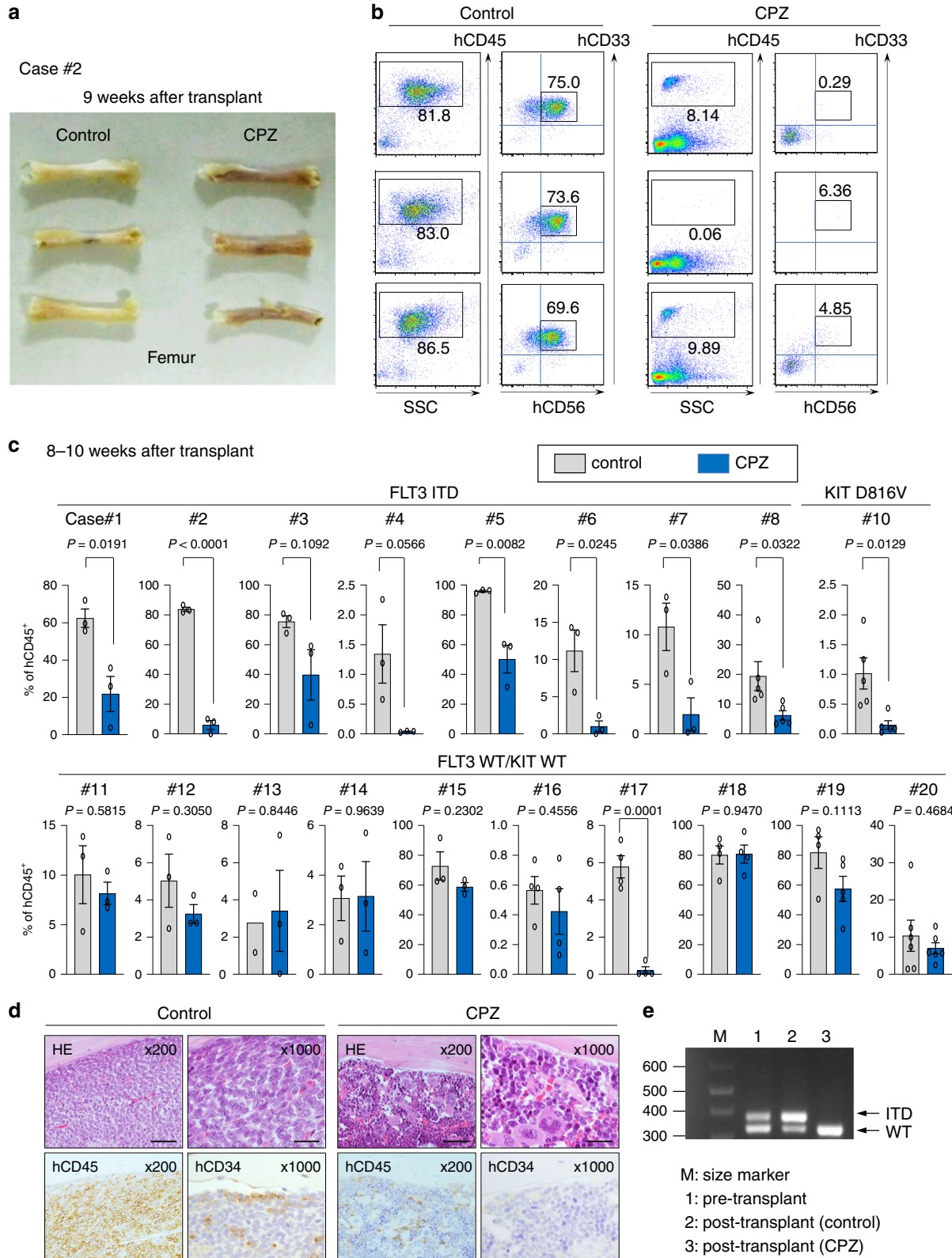

muscarinic acetylcholine receptors[29–31]. Because of its versatile pharmacological profile, CPZ has been used for various diseases such as porphyria, tetanus, severe anxiety, psychotic aggression, refractory hiccups, severe nausea/emesis, insomnia, and severe pruritus[32,33]. In this study, we utilized CPZ in experiments to test its properties as an inhibitor of CCV formation[25] and to our surprise, we observed that CPZ inhibited the growth/survival of

AML cells with MT-RTKs, while it showed minor effects on AML cells with WT-RTKs. To clarify the mechanism underlying anti-AML activities of CPZ, we examined the effects of CPZ on the expression of CALM protein and found that CPZ reduced CALM protein at posttranscriptional levels. Also, we found that the intracellular location of FLT3 ITD and KIT D816V was perturbed by *CALM* iKD or CPZ treatment in MV4-11 and HMC1 cells,

**Fig. 6 CPZ has antileukemic activities against AML cells with MT-RTKs in vivo. a** NOG mice were transplanted with BMMNCs from case #2 with FLT3 ITD[+] AML. These mice were treated with 10 mg/kg of CPZ or normal saline (as a control) from 10 days after transplantation. Nine weeks after transplantation, femurs were collected from CPZ-treated (n = 3, right panel) or control mice (n = 3, left panel). Representative photographs of the collected specimens are shown. **b** BM cells were subjected to flow cytometric analyses. Engrafted cells were detected as a hCD45[+] fraction. In this fraction, engrafted AML cells were further identified as hCD33[+]hCD56[+] cells. **c** NOG mice were transplanted with primary AML cells from patients with FLT3 ITD (cases #1- #8), KIT D816V (case #10), or FLT3 WT/KIT WT (cases #11- #20). Each AML sample was injected into 6–12 mice and these animals were treated with 10 mg/kg of CPZ or normal saline (as a control). Eight to ten weeks after transplantation, mice were euthanized and specimens were collected. The proportion of hCD45[+] cells in the BM or PB was assessed by flow cytometry. Figures depict the mean ± SEM of the %hCD45[+] cells in each group (n = 3 mice per group for case1, 2, 3, 4, 5, 6, 7, 11, 12, 13, 14, 15. n = 4 mice per group for case16, 17,18. n = 5 mice per group for case8, 10, 19. n = 6 mice per group for case20). Two-sided unpaired Student's t test. **d** After transplantation of FLT3 ITD[+] AML cells (from case #2), mice were treated with CPZ or normal saline. Histopathological analysis was performed on BM samples isolated from these mice 9 weeks after transplantation. Representative images are HE staining (upper panel) and immunohistochemical staining to detect engrafted hCD34[+] and hCD45[+] cells (lower panel). Images are representative of three independent experiments. Scale bars, 100 μm at 200 × 20 μm at 1000×. **e** PCR analyses were performed using genomic DNA isolated from BMMNCs before transplantation and from BM samples isolated from CPZ- or normal saline (as a control)-treated mice 9 weeks after transplantation. Images are representative of three independent experiments.

respectively. In addition, we confirmed that a DR antagonist, THIO, and a 5-HTR antagonist, METHIO, showed only marginal growth inhibitory effects on MV4-11 and HMC-1 cells harboring MT-RTKs, while they were highly effective for HL-60, Kasumi, and KG-1 cells with WT-RTKs. We also confirmed that CPZ hardly inhibited the growth of Ba/F3-FLT3 ITD and Ba/F3-KIT D814V cells, in which CALM was already KD. However, CPZ still further suppressed the growth of *CALM* iKD MV4-11 and HMC-1 cells to some extent. Together, these results suggest that, although CPZ selectively targets AML cells with MT-RTKs mainly through CALM protein depletion, its activities to inhibit other molecules such as DR and 5-HTR might partially contribute to full anti-AML activities of CPZ depending on cellular contexts. This speculation is supported by the fact that METHIO and THIO partially inhibited the growth of MV4-11 and HMC-1 cells. Alternatively, because CPZ reduced CALM protein levels in MV4-11 and HMC-1 cells, CPZ might further deplete residual CALM protein escaped from iKD in these cells.

Early epidemiological studies published by the end of the 1960s reported that inmates in mental institutions died from cancer with a substantially lower rate than age- and sex-matched general population[34,35]. Since such a reduction in cancer mortality coincided with the introduction of CPZ in psychiatric clinical practice, some authors proposed a possibility that CPZ might have antitumor properties[36]. In the following decades, a number of studies reported a reduced cancer incidence in patients with psychiatric disorders, especially in those with schizophrenia[37–39]. At that time, a lowered cancer risk in psychiatric patients was attributed to the complex interplay among environmental, dietary, and genetic factors[40]. The potential anticancer effects of antipsychotic drugs have been investigated in several cellular systems. Phenothiazine was first found to induce cell death in various leukemia cell lines, although its molecular mechanisms were not investigated[41]. Subsequent in vitro studies reported CPZ has antitumor activities against colorectal cancer cell lines through activation of the JNK pathway and degradation of SIRT1[42]. More recently, it was reported that endocytosis inhibitors (dynasore, CPZ, and methyl-β-cyclodextrin) have antitumor effects against various cancer cell lines[5,43,44]. However, these effects were cell type-dependent. So, in these studies, the authors speculated that these agents would induce cell death by inhibiting cellular uptake of cell-type specific molecules such as nonviral complexes[43]. In light of the findings described in the present study, it is plausible that these endocytosis inhibitors might also affect intracellular trafficking and/or localization of MT-RTKs and their effects were indeed dependent on the presence or absence of MT-RTKs in tested cell lines.

Both FLT3 ITD and KIT D816V confer poor prognosis on AML patients[45] and a number of preclinical and clinical studies have focused on the development of inhibitors of these MT-RTKs[46,47]. As for FLT3 inhibitors, 1st generation multikinase inhibitors, such as lestaurtinib, sorafenib, and midostaurin, have been investigated, however, clinical efficacies for AML with FLT3 mutations are still limited, and may contribute to adverse effects due to the inhibition of multiple other kinases[11,48]. To overcome these problems, 2nd generation FLT3 inhibitors with selective FLT3 inhibitory activity such as gilteritinib, quizartinib, and crenolanib have been developed[46,47], and gilteritinib was recently approved for relapsed/refractory adult AML patients with mutated FLT3[49]. As for KIT inhibitors, imatinib has been tested in AML with or without KIT D816V because KIT is autophosphorylated in AML cells regardless of the presence of KIT D816V. Although imatinib showed some activities on AML in several clinical studies, its effect was limited[50–52].

Most tyrosine kinase inhibitors (TKIs) block target kinases by binding to their ATP-binding pocket as ATP competitors[53]. However, when a point mutation that interrupts the access of TKI to this pocket occurs, TKI is no longer effective. For example, T315I mutation in the *BCR-ABL* gene is known as a gatekeeper mutation against 1st and 2nd generation BCR-ABL inhibitors[54]. Similarly, T790M mutation in the epidermal growth factor receptor (*EGFR*) gene causes resistance to 1st and 2nd generation EGFR inhibitors[55]. Also, D835Y, D835V, and D835F mutations in the *FLT3* gene confer resistance to quizartinib on AML cells with FLT3 ITD[56]. Furthermore, an intravital imaging study recently showed that once MT-RTKs are internalized by endocytosis, an extremely small amount (at picomolar concentration) of MT-RTKs is sufficient for tumor development[57]. Together, these results indicate that kinase inhibition by ATP-competitive TKIs has a limit to elicit prominent and durable antitumor effects. So, targeting intracellular localization of MT-RTKs as proposed here will be a promising therapeutic approach.

In summary, we here found that CPZ reveals antileukemic activities on AML cells carrying MT-RTKs in vitro and in vivo. Importantly, CPZ exhibited antileukemic activities without inhibiting normal hematopoiesis at a concentration, which is attainable when it is used as an anti-psychiatric drug for psychopathic patients. CPZ also significantly reduced the number of AML initiating cells, proposing its potential to totally eradicate AML. In addition, because anti-AML activities of CPZ were mainly mediated by altering the location of MT-RTKs, our result suggests that blocking compartment-dependent oncogenic signaling of MT-RTKs would be a therapeutic strategy for AML with MT-RTKs. We believe that future clinical trials using CPZ or more potent

analogues are warranted. Furthermore, since MT-RTKs are detectable in a variety of malignancies such as lung cancer with MT-EGFR, hepatocellular carcinoma with MT-MET, and gastric cancer with MT fibroblast growth factor receptor, the findings reported here can be further extended to other malignancies.

## Methods

**Reagents and antibodies (Abs)**. Antibodies are listed in Supplementary Table 2. CPZ and METHIO were purchased from Sigma-Aldrich (St. Louis, MO) and THIO was purchased from Merck Research Laboratories (West Point, PA). These agents were dissolved in distilled water and stored as 10 mM stock solution at 4 °C. Recombinant murine IL-3, SCF, FL, human SCF, FL, and TPO were purchased from Peprotech (Rocky Hill, NJ).

**Cell cultures**. Leukemia cell lines, BALL-1, HL-60, TALL-1, KG-1, THP-1, Kasumi, and K562 were purchased from Japanese Collection of Research Bioresources Cell Bank (National Institutes of Biomedical Innovation, Health and Nutrition, Japan). HMC-1, MV4-11, Ba/F3 and 293T cells were purchased from American Tissue Culture Collection (Manassas, VA). BALL-1, HL-60, TALL-1, KG-1, THP-1, Kasumi, K562, and HMC-1 cells were cultured in RPMI 1640 medium (Gibco, Thermo Fisher Scientific, Waltham, MA) supplemented with 10% fetal calf serum (FCS). Ba/F3 cells were cultured in RPMI 1640 medium supplemented with 10% FCS in the presence of 10 ng/ml IL-3. MV4-11 cells were cultured in Iscove's Modified Dulbecco's Medium (IMDM, Thermo Fisher Scientific) supplemented with 10% FCS. 293T cells were cultured in Dulbecco's modified Eagle's medium (DMEM, Thermo Fisher Scientific) with 10% FCS.

**Retrovirus and lentivirus transduction**. Murine full-length FLT3 WT, FLT3 ITD, KIT WT, and KIT D814V cDNAs gifted from Dr. Mizuki M (Osaka University, Osaka, Japan) were each subcloned into a bicistronic retrovirus vector, pMSCV-IRES-EGFP. These retrovirus vectors were transfected into a packaging cell line 293T containing the expression plasmids for gag and pol. The supernatant was collected 48 h after transfection. Ba/F3 cells were plated onto 3.5 cm dishes coated with fibronectin fragments (Retronectin dish, Takara, Shiga, Japan) and cultured with 1 ml virus supernatant for 72 h. Retrovirus-infected Ba/F3 cells were isolated as GFP-positive cells by FACS Aria (BD Biosciences, Franklin Lakes, NJ). Expression vectors for CALM-specific shRNA and SCR shRNA (a negative control) were generated by cloning these sequences into pGFP-V-RS (OriGene Technologies, Rockville, MD). These shRNA expression vectors were transduced into Ba/F3 clones each expressing FLT3 WT, FLT3 ITD, KIT WT, and KIT D814V by a retrovirus system. shRNA-transfected clones were selected in culture medium containing 1 μg/ml puromycin. The lentiviral vector for a DOX inducible system, Tripz containing shRNA insert specific to CALM (pTRIPZ-shCALM), was custom constructed by GE Healthcare (Cleveland, OH). After transfection of pTRIPZ-shCALM into 293T cells, the supernatants were collected 48 h later. Target cells (MV4-11, HMC-1, and HL-60) were plated onto 3.5 cm dishes coated with fibronectin fragment and cultured with 1 ml virus supernatant for 72 h. Lentivirus-transfected clones were selected in the culture medium containing 1 μg/ml puromycin. To induce the expression of shRNA for CALM, 2 μg/ml DOX (Clontech Laboratories, Mountain View, CA) was added to the culture medium for the indicated times. Levels of target gene mRNA were quantified by qRT-PCR. PCR primers for human CALM (Hs00200318_m1), human KIT (Hs00174029_m1), and human GAPDH (Hs02786624_g1) were purchased from applied biosystems (Thermo Fisher Scientific).

**Isolation of primary AML cells from BM and CB**. BM cells were collected from AML patients, which contained >70% AML cells. Bone marrow mononuclear cells (BMMNCs) were isolated by Ficoll centrifugation and cryopreserved in Cellbanker (Nippon Zenyaku Kogyo Co, Fukushima, Japan) until use. hCD34+ CB cells from a single donor were purchased from Promo Cell (Heidelberg, Germany). Freshly thawed cells were used in all experiments.

**Evaluation of viable cell number**. For proliferation assay, Ba/F3 cells from the indicated clones were seeded at an appropriate cell density ($1 \times 10^4$/ml–$1 \times 10^5$/ml) in RPMI 1640 medium with 10% FCS at 37 °C for 72 h. SCF (100 ng/ml), FL (100 ng/ml), or IL-3 (10 ng/ml) was added according to the purpose of the experiments. In some experiments, 7.0 μM CPZ was added to these cultures. Viable cell number was measured using the Cell Titer Glo Reagent (Promega, Madison, WI), according to the manufacturer's recommendations using an Envision plate reader (Wallac, 1420 ARVO MX-2, Turku, Finland). To analyze the growth of AML samples, BMMNCs or phenotypically sorted AML cells were seeded at a cell density of $1 \times 10^4$/ml–$5 \times 10^4$/ml and cultured in RPMI 1640 with 10% FCS containing 100 ng/ml SCF, 100 ng/ml FL, 100 ng/ml TPO in the presence or absence of 7.0 μM CPZ for the indicated time. The number of viable cells was measured as described above.

**Flow cytometric analysis**. After staining AML cells with the appropriate Abs, the expression of surface molecules was analyzed by a FACS Aria using the BD FACS Diva software (BD Biosciences) and further reanalyzed with Flow Jo software (TreeStar, Ashland, OR). hCD34+CD38− cells were sorted from BMMNCs with FACS Aria. The gating strategy is depicted in Supplementary Fig. 8 and has been applied for all FACS analyses. After staining the cells with a FITC Annexin V Apoptosis Detection Kit II (BD Biosciences), apoptotic cells were detected flow cytometry.

**Immunohistochemical analysis**. Paraformaldehyde-fixed, decalcified, and paraffin-embedded sections were prepared from femur of the mice at the indicated time after transplantation. To retrieve antigens, paraffin sections were incubated with 10 mM citrate buffer at 97 °C for 15 min. After incubation in blocking buffer with 3% hydrogen peroxide, 90% methanol (Wako Pure Chemical, Osaka, Japan) for 15 min, the sections were incubated with the appropriate primary Ab at 4 °C for 12 h and then with the secondary Ab for 30 min. After immunohistochemical staining with 3, 3′-diaminobenzidine (DAB) (Vector Laboratories, CA, USA) for 20 min, images were taken using a clinical microscope instrument (Eclipse-Ci, Nikon, Tokyo, Japan) equipped with a digital camera (DS-Ril, Nikon).

**Immunofluorescence analysis**. Cells were incubated with or without 7.0 μM CPZ for the indicated times, and these cells were rinsed with 0.1 M PBS. Then, these cells were cyto-centrifuged onto coverslips and fixed with 95% methanol for 30 min. After 20 min of incubation in a blocking buffer containing 1% (w/v) bovine serum albumin and 0.1% (v/v) Triton X-100 in PBS, the fixed cells were incubated with the primary Ab and then with the appropriate secondary Ab (each for 45 min). After washing with PBS, the coverslips were mounted on glass slides using 4′,6-diamidino-2-phenylindole (DAPI) Fluoromount-G (Southern Biotech, Birmingham, USA) and observed under a BZ-X710 All-in-One fluorescence microscope (Keyence Corp, Osaka, Japan). Each image showed single sections with an 60× oil immersion objective, adjusted to give the same x, y, and z position in the all channels. The co-localizations were measured via Manders' Colocalization Coefficients using ImageJ software (NIH, Bethesda, MD). These coefficients vary from 0 to 100%, the former corresponding to nonoverlapping images and the latter reflecting complete colocalization between both images.

**Immunoblot and immunoprecipitation**. After washing the cultured cells with PBS, cells were lysed in lysis buffer containing 1% Nonidet P-40 (NP-40) and protease inhibitors for 20 min, and insoluble materials were removed by centrifugation. For immunoprecipitation, protein extracts were incubated with the appropriate Ab and protein A-sepharose beads (Thermo Fisher Scientific) at 4 °C for 6 h and immunoprecipitates were obtained by centrifugation. PDI-positive ER fraction was isolated with an ER Isolation Kit (ER0100, Sigma-Aldrich) following their protocol. Cell lysates (15 μg per each lane) or immunoprecipitated proteins were subjected to SDS–PAGE and electrophoretically transferred onto a polyvinylidene difluoride membrane (Immobilon®, Millipore, Bedford, MA). After blocking residual binding sites on the membrane with TBST blocking buffer (4% nonfat dry milk in Tris-buffered saline-Tween 20, 0.15 M NaCl, 0.01 M Tris-HCl pH 7.4, 0.05% Tween 20), immunoblotting was performed with appropriate primary and secondary Abs. Then, immunoreactive proteins were visualized by an enhanced chemiluminescence kit (LAS4010, GE Healthcare, Cleveland, OH), and densitometric analyses were carried out by Image Quant TL (GE Healthcare). The original images of immunoblots analyses were shown in Supplementary Fig. 9 and 9contnd.

**Detection of FLT3 ITD and KIT D816V**. Genomic DNA was extracted using a Wizard genomic DNA purification kit (Promega). Human FLT3 (both WT and ITD) genes were amplified by PCR as follows. PCR mixture contained 250 ng genomic DNA, 10 pmol human FLT3-ITD primers, 0.2 mM each deoxynucleotide triphosphate, 1× high fidelity PCR buffer, and 1 U of platinum Taq DNA polymerase high fidelity (Thermo Fisher Scientific). After denaturation step at 94 °C for 3 min, PCR consisting of denaturation (94 °C for 30 s) annealing (56 °C for 1 min), and extension steps (72 °C for 2 min) was repeated until 35 cycles, followed by final extension step at 72 °C for 10 min on a GeneAmp PCR system 9700 (Thermo Fisher Scientific). After electrophoresis on 2% agarose gels, PCR products were visualized by ethidium bromide staining. Human KIT gene was initially amplified by 40 cycles of PCR with human KIT-D816V primers as follows: denaturation (95 °C for 30 s), annealing, (61 °C for 25 s), and extension (72 °C for 25 s). After visual confirmation of amplification, the PCR product (3 ng), a primer set as described above (2 pmol/μL), BigDye terminator ver.3.1 cycle sequencing mix, and sequencing buffer were mixed and subjected to further PCR consisting of 30 cycles of denaturation (96 °C for 10 s), annealing (50 °C for 5 s), and extension (60 °C for 4 min). Sequence analysis was performed using genetic analyzer 3100 (Thermo Fisher Scientific). The sequences of the PCR primers used in this study were:

Human FLT3-ITD forward: 5′-GCAATTTAGGTATGAAAGCCAGC-3′
Human FLT3-ITD reverse: 5′-CTTTCAGCATTTTGACGGCAACC-3′
Human KIT-D816V forward: 5′-CCTCCAACCTAATAGTGTATTCACAG-3′
Human KIT-D816V reverse: 5′-ATGTGTGATATCCCTAGACAGGAT-3′

**In vivo mouse models**. For transplantation of Ba/F3 FLT3 ITD/Mock or/*CALM* KD cells ($5 \times 10^3$ cells) were suspended in 100 µl PBS and injected via tail vein into 6–8-week old female BALB/C mice. Animals were euthanized 2–3 weeks after transplantation and subjected to analyses. The number of engrafted cells in BM or PB was counted by flow cytometric analysis as GFP-positive cells. To transplant Ba/F3 KIT D814V/Mock and/*CALM* KD cells (each $2 \times 10^6$ cells/100 µl PBS) were subcutaneously injected into two sites in the posterior flank of 2.5 Gy irradiated 6–8-week old nude mice (BALB/cAJcl-nu/nu mice purchased from CLEA Japan, Inc., Tokyo, Japan). Tumor development was visually monitored after injection. The survival of these mice was compared by a Kaplan–Meier analysis.

For xenotransplantation of primary AML cells, 6–8-week old female NOD/Scid/IL2Rγ-KO (NOG) mice purchased from Central Institute for Experimental Animals (Kawasaki, Kanagawa, Japan) were irradiated with 2.5 Gy, followed by the intravenous injection of BMMNCs cells from AML patients ($1 \times 10^6$ BMMNCs suspended in 100 µl PBS) within 2 h. Animals were then treated with 10 mg/kg CPZ or NS intraperitoneally once-daily from the indicated days after transplantation. After 8–10 weeks from transplantation, mice were euthanized and samples were collected for analysis. BM cells were subjected to flow cytometric analysis and femur sections were prepared for histological analysis. For xenotransplantation of normal human HSC cells, $1 \times 10^4$ CD34$^+$ CB cells were transplanted into NOG mice as described above. From day 10th after transplantation, these mice were treated with CPZ on the same protocol as described above. BM cells were harvested 8–10 weeks after transplantation and analyzed by flow cytometry. All mice were bred and maintained under specific-pathogen-free conditions at the animal facilities of Kindai University, and animals are kept in groups of 5–6 in individually ventilated caging system and provided continuously with sterile water and chow pellets. The intra-cage temperature was $23 \pm 1\ °C$ and the relative humidity $50 \pm 10\%$. A 12:12 Dark–Light cycle was operating in the room.'

**Statistics and reproducibility**. Statistical analysis was performed using the Graph Pad prism 6 software package (San Diego, CA, USA). At least three independent cell samples or mice were included for statistical analysis. Statistical analysis was performed with the Student $t$ test. One-way analysis of variance (ANOVA) with Bonferroni's post-hoc test was used for comparisons among three or more groups. Survival curves of mice transplanted with various Ba/F3 clones (at least $n = 10$ per group sample size) were analyzed by Kaplan–Meier survival estimate using a Wilcoxon test. Error bars indicate the standard error (SE) of the mean. $P$ values < 0.05 were considered statistically significant. All the experimental findings were reliably reproduced. The exact numbers are clearly indicated in the figure legends.

**Ethical statements**. AML patient samples with or without FLT3 ITD or KIT mutations were obtained after written informed consent following institutional guidelines of Kindai University Faculty of Medicine (Authorization Number:24-017, −018) per Declaration of Helsinki principles. All experiments were conducted after getting the approval from Ethics Committee of Kindai University Faculty of Medicine (Approval ID 29-072). All animal experiments were conducted after getting the approval from the Animal Care and Use Committee for Kindai University (Approval ID 06-13).

**Reporting summary**. Further information on research design is available in the Nature Research Reporting Summary linked to this article.

## Data availability

The source data underlying Figs. 1a–c, 2a–e, 3a-e, 4c, f, 5a–d, 6c and Supplementary Figs are provided as a Source Data file. All relevant data are available from the authors upon reasonable request. Source data are provided with this paper.

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

## Acknowledgements

Technical help and valuable discussion of Ms. Furukawa K, Mr. Mizuguchi N, Mr. Horiuchi Y, Mr. Kurashimo S (Kindai University Faculty of Medicine), and Prof. Yoshimori T (Department of Genetics, Osaka University Graduate School of Medicine) is acknowledged. This work was supported by Grants-in-aid for Scientific Research (C) (15K09461) from the Japan Society for the Promotion of Science. The funders had no role in study design, data collection and analysis, decision to publish, or preparation of the paper. (http://kaken.nii.ac.jp/).

## Author contributions

H.T. and I.M. designed and supervised research. S.R., M.S., T.K, A.T. and H.T. analyzed the data. S.R. and H.T. provided disease-specific analysis. S.R. and H.T. performed statistical analysis. S.R., M.S., A.T., L.E. and H.T. prepared figures and tables. S.R., H.T., L.E., T.W. and I.M. wrote the paper. T.Y., K.O., T.W. and Y.K. revised the paper.

## Competing interests

The authors declare no competing interests.
