## [Peer Review File · Nature Communications]

Reviewers' comments:

Reviewer #1 (Remarks to the Author):

Matsumura and colleagues presented a paper where the intracellular trafficking is studied as a possible therapeutic target for AML. The preliminary data that served the basis for this paper is a clinical case. An AML patient was treated with chlorpromazine and achieved remission from the AML disease. Although these findings are extremely interesting for the field (there is an unmet need for AML), the experimental design can be greatly improved.

- Chlorpromazine (CPZ). Although authors claimed that this compound affects clathrin-mediated intracellular trafficking, CPZ has been well described as an antagonist of dopamine receptor, serotonin receptor and histamine receptor. Both, dopamine receptor and serotonin receptor has been previously described as therapeutic targets for leukemic stem cells, using AML as a model (Sachlos et al. Cell 2012; Etxabe et al Leukemia 2017). Therefore:

1. The novelty is highly compromised
2. The MOA studied is not clear as the well-established and well reported has not been discarded.

- Dopamine receptor and serotonin receptor. Authors used CPZ, a well described Dopamine and serotonin receptor antagonists. Both receptors are ligand-dependent endocytosed in a clathrin-dependent manner. Therefore, the effect seen could be due to the inhibition of these receptors. On the other side, based on the literature available, the effect of CPZ observed can be perfectly explained by the inhibition of dopamine receptor or serotonin receptor.

- AML stem cells (LSC). The authors claimed that CPZ have effect on LSCs (something expected based on the literature available that have already demonstrated this) based on data generated on CD34+CD38- AML cells. However, it has been extensively demonstrated that LSC can not be phenotypically identified.

- Secondary transplants. Functionality of LSCs should be assayed in secondary transplantation assays.

- Due to the importance of the clinical case for the development of this paper, I will combine it with the experimental data.

Regardless, as CPZ has been widely reported as a dopamine/serotonin receptor antagonist and both receptors are well known as AML targets, the novelty of the paper is highly compromised. Moreover, the effect of intracellular trafficking has not been proven experimentally in this work.

Reviewer #2 (Remarks to the Author):

The manuscript by Rai et al. provides evidence that an anti-psychotic drug chlorpromazine (CPZ) could be considered a therapeutic option to treat acute myeloid leukemia (AML) harboring activating mutations in RTKs. The authors provide a comprehensive set of data, obtained both in vitro and in vivo, showing that CPZ inhibits the growth of AML cells with mutated RTKs, such as FLT3 ITD and KIT D816V, but not of cells that rely on other oncogenic mechanisms. At the molecular level, among other activities, CPZ inhibits clathrin-mediated endocytosis. The authors further show that the knockdown of CALM, a protein involved in the formation of clathrin-coated vesicles (CCVs), exerts similar effects as CPZ, i.e. specifically inhibits proliferation of AML cells with the FLT3 ITD and KIT D816V mutations. Based on the similarity of the phenotypes of CPZ application and CALM depletion, as well as some additional data, the authors propose that the inhibition of intracellular trafficking of mutated RTKs leads to reduced cell growth. The final results

obtained using CPZ in patient-derived AML cells cultured in vitro and xenografted in mice are supported by observations in the attached case report on a human patient and are clearly of high relevance for the clinic.

Overall, this study very convincingly documents a selective inhibitory activity of CPZ towards a subset of AML cells in vitro and in vivo (and in one human patient). This is a novel finding that may be of great significance for the emerging field of personalized medicine. However, the major problem of the present manuscript is that it fails to identify a molecular mechanism underlying these effects. The main conclusion given in the title - that it is the blocking of intracellular trafficking of mutant RTKs that kills AML cells - is not well supported by the data and remains not proven. The study lacks sufficient mechanistic data that would properly reveal the biological processes affected by CPZ in these tumor cells. While the inhibitory effect of CPZ on the growth of AML cells with mutated RTKs is undeniable, the authors do not provide any proof that it is due to the CPZ action on intracellular trafficking (and if yes, in what way the receptor trafficking is affected). The scarce mechanistic insight presented in this manuscript is based on similar effects on cell proliferation and signaling caused by CALM depletion or CPZ treatment in a mouse pro-B cell line Ba/F3. Further results showing an apparently altered intracellular distribution of the mutant RTKs in MEF cells treated with the inhibitor or lacking CALM protein are not convincing (see below). On another note, it is well established that CPZ blocks clathrin-mediated endocytosis (and possibly other clathrin-mediated trafficking routes) but the exact molecular mechanism of its action is not clear (see e.g. Daniel et al *Traffic*. 2015 Jun;16(6):635-54. doi: 10.1111/tra.12272). In fact, it may not be related to the formation of CCVs, as the authors repeatedly write. Moreover, while referring to the known activity of CPZ on membrane trafficking, the authors did not mention that chlorpromazine has many other effects, for example, on the activity of enzymes like phosphatidic acid phosphorylase, protein kinase C or calcium-calmodulin-dependent enzymes (as described in the review by von Kleist and Haucke, 2012, cited by the authors). Knowing that, the dissection of the molecular mode of CPZ action on leukemic cells should be performed more thoroughly.

In essence, the translational part of the study is mostly solid and well controlled/conducted, while the part on the molecular mechanism is - unfortunately - insufficient and flawed.

Major issues:

1. The immunofluorescence data on the RTK intracellular localization in MEFs (Fig. 3A, S3) are not convincing. The staining quality is poor, also for the well-established markers, thus any redistribution is not obvious. Such data (e.g. the degree of colocalization) should be quantified from several independent experiments. Scale bars are missing.
2. The authors refrain from analyzing the receptor localization in myeloid-derived cells, due to their small size. However, the observations done in MEF cells, although important, cannot be extrapolated to the situation in leukemic cells, in which intracellular RTK trafficking might be regulated in a different manner than in MEFs. Hence, the authors should examine whether and how CPZ alters the intracellular localization of the mutated RTKs in leukemic cells. Although, as pointed out by the authors, immunofluorescence analysis might not be accurate, gradient analysis should provide the required information even in cells with small cytoplasmic volume. Moreover, the type of the gradient used (sedimentation, floatation?) should be clearly specified.
3. Oncogenic Kit has been shown to activate AKT from endosomes, while STAT5 from the ER (Obata et al. 2014 *Nat Commun*. 2014 Dec 10;5:5715. doi: 10.1038/ncomms6715). Here the authors did not provide data regarding the effect of CALM depletion or CPZ treatment on STAT5 phosphorylation in KIT D814V cells. In the gradient experiment (Fig. 3B) the mutated KIT receptor is redistributed upon CPZ from the Golgi fractions to some lighter fractions which, as judged by the PDI protein abundance, should be the ER. As KITD814V is immunolocalized only with respect to syntaxin-6 and not to PDI (or other ER markers), as it is the case for FLT3 ITD, the authors did not rule out that upon inhibition of intracellular trafficking KIT D814V does not accumulate on the ER and induce STAT5 overactivation. As STAT5 activation under some circumstances may be beneficial for tumor cells, this aspect should be addressed.

4. To address whether an anti-proliferative effect of CPZ is due to its impact on intracellular trafficking, the authors should answer the following questions. Does depletion of CALM (or other trafficking regulator/s) affect signaling or proliferation of RTK-mutated AML cells? Does depletion of CALM change the intracellular localization of mutated RTKs? Is the reduced signaling and/or proliferation due to CALM depletion further inhibited by CPZ in Ba/F3 cells or, more importantly, how do CALM-silenced AML cells respond to CPZ?

5. The authors do not discuss what the fate of FLT3 ITD receptor is when it is redistributed after CPZ from the ER. Does it accumulate in some Golgi subcompartment or elsewhere? Why do the two mutated receptors behave differently upon CPZ? These aspects, as well as the other above mentioned issues (after addressing them experimentally), should be interpreted in the discussion section which in the current version of the manuscript puts little focus on the molecular consequences of CPZ treatment.

6. The text contains many wrong pieces of information about intracellular trafficking. For example: Introduction, the second sentence: "Receptor tyrosine kinase (RTK) is a high-affinity transmembrane protein, which is internalized by CME". Not all RTKs are internalized by CME and in most known cases CME is only one of possible internalization routes.

Introduction, line 64: "a part of CCVs containing WT RTKs is sorted back to the plasma membrane via recycling endosomes, while the remaining part is transported to late endosomes and consequently degraded at lysosomes". CCVs are not sorted back via recycling endosomes or degraded at lysosomes. It is the membrane cargo that is sorted. Clathrin is removed from vesicles before fusing with endosomes and thus CCVs cease to exist once fused with endosomes.

Discussion, the first sentence: "CALM functions in the initial step of CME by facilitating the formation of CCV, which is primarily found at the trans-Golgi network". This statement is generally unclear but if the authors talk about CCVs mediating endocytosis, then these vesicles are formed at the plasma membrane, not at the TGN (where CCVs have other non-endocytic functions which are not discussed by the authors at all).

Overall, it is not clear why in the introduction and discussion the authors refer mainly to the effects of CPZ on endocytosis, while the phenotypes they observe in AML cells are more likely related to other clathrin-mediated trafficking routes, e.g. initiated at the TGN.

Minor issues:

- The immunoblots showing changes in signaling should be quantified from several independent experiments (e.g. Fig. 1C, S2).

- p. 3, line 69: The cited references (8 and 9) are outdated and do not include information about the mutated Kit receptor localization on the ER which should be cited (Obata et al. 2014 Nat Commun. 2014 Dec 10;5:5715. doi: 10.1038/ncomms6715).

- The description of Fig. 5 and 6 in the text is unclear. The authors do not explain which types of cells are CD45-positive or CD34-positive. They use interchangeably CD45 or LCA without explaining that it is the same marker.

- Generally, the language (grammar/spelling) should be improved, below only a handful of more numerous examples:

- p. 3, line 71: missing "is" at the end of the line

- p. 3, line 79: missing "is" before "expressed"

- p. 4, line 90: missing "as" before "oncogene"

- p. 10-11, lines 288-289 Incorrect sentence. Should be: "... and even in cases of effective FLT3 inhibition, the effects were transient ..."

- Table S1 should have some legend/explanation

- Fig. 4D Misplaced dashed line in #5?

Reviewers' comments:

Reviewer #1 (Remarks to the Author):

Matsumura and colleagues presented a paper where the intracellular trafficking is studied as a possible therapeutic target for AML. The preliminary data that served the basis for this paper is a clinical case. An AML patient was treated with chlorpromazine and achieved remission from the AML disease. Although these findings are extremely interesting for the field (there is an unmet need for AML), the experimental design can be greatly improved.

- Chlorpromazine (CPZ). Although authors claimed that this compound affects clathrin-mediated intracellular trafficking, CPZ has been well described as an antagonist of dopamine receptor, serotonin receptor and histamine receptor. Both, dopamine receptor and serotonin receptor has been previously described as therapeutic targets for leukemic stem cells, using AML as a model (Sachlos et al. Cell 2012; Etxabe et al Leukemia 2017). Therefore:

1. The novelty is highly compromised.

Response:

We thank the reviewer for important and critical comments for our findings. To confirm that anti-AML activities of CPZ were mediated by altering the intracellular localization of mutated- (MT-) RTKs and not by previously proposed mechanisms (inhibition of serotonin(5-HT) and/or dopamine signals), we have performed several additional experiments.

At first, we compared the effects of CPZ with those of 5-HT and dopamine receptor antagonists (Methiothepin, METHIO and Thioridazine, THIO, respectively) on the growth and survival of several leukemia cell lines with wild type- (WT-) or MT-RTKs. We verified that HL-60 cells with WT-RTKs (FLT WT/ KIT WT) (which express high levels of 5-HT and dopamine receptors) are highly sensitive to METHIO and THIO, but less sensitive to CPZ compared to MV4-11 cells (with FLT3 ITD) and HMC-1 cells (with KIT D814V). Conversely, the growth of MV4-11 and HMC-1 cells was only partly (about 10-30%) inhibited by METHIO or THIO.

(We have added these findings as Fig. 3a.b.in Result sections as follows:

Result section p.6 Line 172~ in the revised paper.)

In a previous version, we showed that both CPZ treatment and CLAM knock down (KD) drastically inhibited MT-RTK-dependent growth of Ba/F3 cells in cytokine deprived

conditions, while it hardly affected ligand (FLT3 or SCF)-activated WT-RTK-dependent growth of Ba/F3 cells. To further clarify the roles of CALM in the growth of AML cells, we knocked down CALM in MV4-11, HMC-1, and HL-60 cells with a doxycycline (DOX) inducible system. At first, we confirmed that induced CALM shRNA reduced the expression of CALM protein to less than 20% compared with SCR RNA after 72-h DOX treatment (shown as Fig. S1a). Induced CALM KD (iKD) didn't influence the growth of HL-60 cells, while it impaired the growth of MV4-11 and HMC-1 cells, indicating that CALM plays an important role in the growth AML cells with MT-RTKs.

(We have added these findings as Fig. 1b in Result sections as follows: Result section p.5 Line 126~ in the revised paper.)

In addition, we found that CPZ treatment reduced CALM protein levels in MV4-11 and HMC-1 cells, while it increased clathrin heavy chain protein levels in both cell lines.

(We have added these findings as Fig. 3c in Result sections as follows: Result section p.7 Line 185~ in the revised paper.)

Moreover, we confirmed that CPZ didn't further augment growth inhibitory effects of CALM KD in Ba/F3-FLT3 ITD and Ba/F3-KIT D814V cells (shown as Fig. 2b).

However, we also found that CPZ still partly inhibited the growth of CALM iKD MV4-11 and HMC-1 cells (shown as Fig. 3e).

So, we speculate that, although CPZ selectively targets AML cells with MT-RTKs mainly through CALM depletion, inhibition of other molecules such as DR and 5-HTR would, to some extent, contribute to full anti-AML activities of CPZ depending on cellular contexts. Alternatively, because CPZ reduced CALM protein levels in MV4-11 and HMC-1 cells, CPZ might further deplete residual CALM protein escaped from KD in these cells.

We have added these results in the Result and Discussion section as follows:

Result section p.6 Line 148~ in the revised paper (Fig. 2b).

Result section p.7 Line 191~ in the revised paper (Fig. 3e).

Discussion section p.13 Line 368~ in the revised paper.

2. The MOA studied is not clear as the well-established and well reported has not been discarded.

Response:

As described above, we found that CPZ treatment reduced CALM protein levels in MV4-11 and HMC-1 cells, while it increased clathrin heavy chain protein levels in both cell lines.

Because CALM mRNA expression levels were increased by CPZ in HMC-1 cells, we considered that CPZ suppressed CALM protein levels at post-transcriptional levels. (We have added these findings as Fig. 3d in Result sections as follows: Result section p.7 Line 185~ in the revised paper.)

To further clarify the effects of CPZ in MT-RTKs, we performed immunoelectron microscopy studies. Under untreated conditions, MT-RTKs (FLT3 ITD in MV4-11 cells and KIT D816V in HMC-1 cells) were detectable in vesicle-like structures, which apparently accumulated at ER and endolysosomes, respectively. In contrast, when CALM was knocked down or these cells were treated with CPZ, vesicle-like structures almost disappeared from ER or endolysosomes, and both FLT3 ITD and KIT D816V spread in cytoplasm with a diffuse pattern.

(We have added these findings as Fig. 4a,4b in Result sections as follows: Result section p.8 Line 224~ in the revised paper.)

Next, we conducted immunoblot analyses using ER and endolysosome proteins isolated from MV4-11 and HMC-1 cells, respectively. We isolated ER with an ER Isolation Kit (ER0100, Sigma-Aldrich) and endolysosome protein was immunoprecipitated by the anti-LAMP1 Ab from total cellular lysates. Without any treatment, FLT3 ITD was detected in the ER protein and KIT D816V in the endolysosome protein. In contrast, when CALM was knocked down or these cells were treated with CPZ, both FLT3 ITD and KIT D816V disappeared from ER and endolysosomes, respectively, which resulted in the reduced phosphorylation of MT-RTKs themselves and their downstream molecules such as STAT5 and Akt.

(We have added these findings as Fig. 5 in Result sections as follows: Result section p.8 Line 229~ in the revised paper.)

Together, these novel findings suggest that CPZ disrupts the intracellular localization of MT RTKs at ER/endolysosomes (confirmed by immunoelectron microscope) and interrupts their association (confirmed by immunoblot analyses) primarily through down regulation of CALM protein, thereby inhibiting compartment-dependent leukemogenic signals from MT-RTKs.

- **Dopamine receptor and serotonin receptor.** Authors used CPZ, a well described dopamine and serotonin receptor antagonists. Both receptors are ligand-dependent endocytosed in a clathrin-dependent manner. Therefore, the effect seen could be due to the inhibition of these receptors. On the other side, based on the literature available, the effect of CPZ observed can be perfectly explained by the inhibition of dopamine receptor or serotonin receptor.

Response:

As described above, we compared the effects of CPZ, METHIO, and THIO on the growth and survival of several leukemia cell lines with WT- or MT-RTKs.

Although both THIO and METHIO effectively inhibited the growth and survival of AML cell lines with WT-RTKs (FLT WT/ KIT WT), these antagonists showed poor inhibitory effects against MV4-11 and HMC-1 cells lines with MT-RTKs compared to CPZ. Also, CPZ didn't further suppress the growth of CALM KD Ba/F3-FLT3 ITD and Ba/F3-KIT D814V. However, CPZ still partly inhibited the growth of CALM iKD MV4-11 and HMC-1 cells. So, we speculate that, although CPZ selectively targets AML cells with MT-RTKs mainly through CALM depletion, inhibition of other molecules such as DR and 5-HTR would, to some extent, contribute to full anti-AML activities of CPZ depending on cellular contexts. Alternatively, CPZ might further deplete residual CALM protein escaped from KD in these cells.

(We have added these findings in Discussion sections as follows: Discussion section p.13 Line 371~ in the revised paper.)

- **AML stem cells (LSC).** The authors claimed that CPZ have effect on LSCs (something expected based on the literature available that have already demonstrated this) based on data generated on CD34⁺CD38⁻ AML cells. However, it has been extensively demonstrated that LSC cannot be phenotypically identified.

- **Secondary transplants.** Functionality of LSCs should be assayed in secondary transplantation assays.

Response:

As pointed out by the reviewer, AML LSCs are not restricted to CD34⁺CD38⁻ phenotype but can be found in a wide variety of phenotypic compartments (e.g., CD34⁺CD38⁺, CD34⁻ fractions and so on), depending on AML samples and experimental models.

To examine the potential of CD34⁺CD38⁻ AML cells as LSCs in our study design, we performed secondary transplantation using CD34⁺CD38⁻ and CD34⁺CD38⁺ cells isolated from mice that developed AML after the 1st transplantation (from cases #2, #3, #5 and #8). Eight weeks after transplantation, CD34⁺CD38⁻ cells were capable of developing AML in the secondary recipient mice, while CD34⁺CD38⁺ cells were not, indicating that CD34⁺CD38⁻ cells utilized in our study have a potential to act as AML LSCs.

However, not all of CD34⁺CD38⁻ cells have AML initiating activities. So, we have modified the explanation of CD34⁺CD38⁻ AML cells from “AML LSCs” to “a CD34⁺CD38⁻ AML fraction containing AML initiating cells”.

(We have shown these results as Fig. S3 and described in Result section as follows: Result section from p.6 Line 162~ in the revised paper.)

- Due to the importance of the clinical case for the development of this paper, I will combine it with the experimental data.

Response:

We appreciate an important suggestion from the reviewer, which will greatly strengthen our hypothesis. We have added the experimental data using AML cells from this patient (as Case #9). As observed in other AML samples with MT-RTKs (Fig. 6), AML cells from this patient were highly sensitive to CPZ in vitro and in vivo. We have shown these data as Fig. 7 a, b, c, d. We have also expanded the discussion about clinical implications of CPZ.

Result section from p.11 Line 305~ in the revised paper (Fig.7)

Discussion section, from page 15, Line 432~in the revised paper.

NCOMMS-18-25851A

Reviewer #2 (Remarks to the Author):

The manuscript by Rai et al. provides evidence that an anti-psychotic drug

chlorpromazine (CPZ) could be considered a therapeutic option to treat acute myeloid leukemia (AML) harboring activating mutations in RTKs. The authors provide a comprehensive set of data, obtained both in vitro and in vivo, showing that CPZ inhibits the growth of AML cells with mutated RTKs, such as FLT3 ITD and KIT D816V, but not of cells that rely on other oncogenic mechanisms. At the molecular level, among other activities, CPZ inhibits clathrin-mediated endocytosis. The authors further show that the knockdown of CALM, a protein involved in the formation of clathrin-coated vesicles (CCVs), exerts similar effects as CPZ, i.e. specifically inhibits proliferation of AML cells with the FLT3 ITD and KIT D816V mutations. Based on the similarity of the phenotypes of CPZ application and CALM depletion, as well as some additional data, the authors propose that the inhibition of intracellular trafficking of mutated RTKs leads to reduced cell growth. The final results obtained using CPZ in patient-derived AML cells cultured in vitro and xenografted in mice are supported by observations in the attached case report on a human patient and are clearly of high relevance for the clinic.

Overall, this study very convincingly documents a selective inhibitory activity of CPZ towards a subset of AML cells in vitro and in vivo (and in one human patient). This is a novel finding that may be of great significance for the emerging field of personalized medicine. However, the major problem of the present manuscript is that it fails to identify a molecular mechanism underlying these effects. The main conclusion given in the title - that it is the blocking of intracellular trafficking of mutant RTKs that kills AML cells - is not well supported by the data and remains not proven. The study lacks sufficient mechanistic data that would properly reveal the biological processes affected by CPZ in these tumor cells. While the inhibitory effect of CPZ on the growth of AML cells with mutated RTKs is undeniable, the authors do not provide any proof that it is due to the CPZ action on intracellular trafficking (and if yes, in what way the receptor trafficking is affected). The scarce mechanistic insight presented in this manuscript is based on similar effects on cell proliferation and signaling caused by CALM depletion or CPZ treatment in a mouse pro-B cell line Ba/F3. Further results showing an apparently altered intracellular distribution of the mutant RTKs in MEF cells treated with the inhibitor or lacking CALM protein are not convincing (see below). On another note, it is well established that CPZ blocks clathrin-mediated endocytosis (and possibly

other clathrin-mediated trafficking routes) but the exact molecular mechanism of its action is not clear (see e.g. Daniel et al Traffic. 2015 Jun;16(6):635-54. doi: 10.1111/tra.12272). In fact, **it may not be related to the formation of CCVs**, as the authors repeatedly write. Moreover, while referring to the known activity of CPZ on membrane trafficking, the authors did not mention that CPZ has many other effects, for example, on the activity of enzymes like phosphatidic acid phosphorylase, protein kinase C or calcium–calmodulin-dependent enzymes (as described in the review by von Kleist and Haucke, 2012, cited by the authors). Knowing that, **the dissection of the molecular mode of CPZ action on leukemic cells should be performed more thoroughly.**

In essence, the translational part of the study is mostly solid and well controlled/conducted, while the part on the molecular mechanism is - unfortunately - insufficient and flawed.

Major issues:

1. The immunofluorescence data on the RTK intracellular localization in MEFs (Fig. 3A, S3) are not convincing. The staining quality is poor, also for the well-established markers, thus any redistribution is not obvious. Such data (e.g. the degree of colocalization) should be quantified from several independent experiments. Scale bars are missing.

Response:

We thank the reviewer for pointing out several critical issues about our manuscript. In order to make them clearer, we quantified the proportion of FLT3 ITD and KIT D814V that were colocalized with ER and Golgi in MEFs treated with or without CPZ, respectively. As a result, CPZ treatment significantly reduced the degree of colocalization of FLT3 ITD/PDI and KIT D816V/syntaxin (shown as Fig. S4b). Also, we have added scale bars in each picture.

To answer the reviewer's comment, we knocked down CALM in an inducible manner (CALM iKD) in various AML cell lines such as MV4-11 (with FLT3 ITD), HMC-1 (with KIT D816V), and HL-60 (with WT FLT3 / WT KIT) using a doxycycline (DOX) inducible system. CALM iKD didn't influence the growth of HL-60 cells, while it

impaired the growth of MV4-11 and HMC-1 cells, indicating that CALM plays an important role in the growth AML cells with MT-RTKs.

(We have added these findings as Fig. 1b in Result sections as follows: Result section p.5 Line 126~ in the revised paper.)

Using these cells, we performed immunoelectron microscope studies. Under untreated conditions, MT-RTKs (FLT3 ITD in MV4-11 cells and KIT D816V in HMC-1 cells) were detectable in vesicle-like structures, which apparently accumulated at ER and endolysosomes, respectively (shown as Fig. 4a, 4b upper panels). In contrast, when CALM was knocked down or these cells were treated with CPZ, vesicle-like structures almost disappeared from ER or endolysosomes, and both FLT3 ITD and KIT D816V spread in cytoplasm with a diffuse pattern.

(We have added these findings as Fig. 4a,4b in Result sections as follows: Result section p.8 Line 224~ in the revised paper.)

Next, we conducted immunoblot analyses using ER and endolysosome proteins isolated from MV4-11 and HMC-1 cells, respectively. We isolated ER with an ER Isolation Kit (ER0100, Sigma-Aldrich) and endolysosome protein was immunoprecipitated by the anti-LAMP1 Ab from whole cellular lysates. Without any treatment, FLT3-ITD was detected in the ER protein (confirmed by the anti-PD1 blot) and KIT D816V in the endolysosome protein. In contrast, when CALM was knocked down or these cells were treated with CPZ, both FLT3 ITD and KIT D816V were scarcely localized at ER and endolysosomes, respectively, which resulted in the reduced phosphorylation of MT-RTKs themselves and their downstream molecules such as STAT5 and Akt. We have shown these results as Fig. 5.

(We have added these findings as Fig. 5 in Result sections as follows: Result section p.8 Line 229~ in the revised paper.)

Together, these novel findings suggest that CPZ disrupts the intracellular localization of MT RTKs at ER/endolysosomes (confirmed by immunoelectron microscope) and interrupts their association (confirmed by immunoblot analyses), thereby inhibiting compartment-dependent leukemogenic signals from MT-RTKs.

2. The authors refrain from analyzing the receptor localization in myeloid-derived cells, due to their small size. However, the observations done in MEF cells, although important, cannot be extrapolated to the situation in leukemic cells, in which

intracellular RTK trafficking might be regulated in a different manner than in MEFs. Hence, the authors should examine whether and how CPZ alters the intracellular localization of the mutated RTKs in leukemic cells. Although, as pointed out by the authors, immunofluorescence analysis might not be accurate, gradient analysis should provide the required information even in cells with small cytoplasmic volume. Moreover, the type of the gradient used (sedimentation, floatation?) should be clearly specified.

Response:

We appreciate important suggestions. We used OptiPrep, the optimum density gradient medium to isolate several organelles as each floated fraction. We described this information in Material and Methods.

To answer the reviewer's comment, we knocked down (KD) CALM in an inducible manner in MV4-11 and HMC-1 cells with a DOX inducible system as described above. Immunoelectron microscope and immunoblot analyses showed that CALM iKD or CPZ altered intracellular location of FLT3 ITD and KIT D816V, the details of these results are described above.

3. Oncogenic Kit has been shown to activate AKT from endosomes, while STAT5 from the ER (Obata et al. 2014 Nat Commun. 2014 Dec 10;5:5715. doi: 10.1038/ncomms6715). Here the authors did not provide data regarding the effect of CALM depletion or CPZ treatment on STAT5 phosphorylation in KIT D814V cells. In the gradient experiment (Fig. 3B) the mutated KIT receptor is redistributed upon CPZ from the Golgi fractions to some lighter fractions which, as judged by the PDI protein abundance, should be the ER. As KITD814V is immunolocalized only with respect to syntaxin-6 and not to PDI (or other ER markers), as it is the case for FLT3 ITD, the authors did not rule out that upon inhibition of intracellular trafficking KIT D814V does not accumulate on the ER and induce STAT5 overactivation. As STAT5 activation under some circumstances may be beneficial for tumor cells, this aspect should be addressed.

Response:

As pointed out by the reviewer, KITD814V was redistributed from Golgi fractions to some lighter fractions supposed to be ER fractions in the gradient experiment using MEF cells (shown as Fig. S4c in the revised supplementary materials), where it might

activate STAT5 in excess. To examine this possibility, we assessed the influence of CALM depletion on STAT5 phosphorylation in HMC-1 cells. However, contrast to this hypothesis, phosphorylation of STAT5 was reduced by CALM iKD in HMC-1 cells (shown as Fig. 1c). Because CALM iKD led to the disappearance of vesicle-like structures containing KIT D816V from endolysosomes and KIT D816V spread in cytoplasm with a diffuse pattern in HMC-1 cells (shown as Fig. 4b), we speculated that CALM is essential not only for the intracellular trafficking of KIT D816V but also for its signal transduction.

(We have added these findings as Fig. 1c in Result sections as follows: Result section p. 5 Line 134~ in the revised paper.)

4. To address whether an anti-proliferative effect of CPZ is due to its impact on intracellular trafficking, the authors should answer the following questions.

Does depletion of CALM (or other trafficking regulator/s) affect signaling or proliferation of RTK-mutated AML cells?

Response:

We previously examined the effects of CPZ on the growth of Ba/F3-FLT3 ITD and Ba/F3-KIT D814V cells, because the growth of these cells is essentially dependent on MT-RTKs. However, as suggested by the reviewer, it is important to confirm these findings in AML cells. So, we examined the role of CALM in AML cell lines such as HL-60 (with WT-RTK), MV4-11 (with FLT3 ITD) and HMC-1 (with KIT D816V) using a DOX-inducible shRNA system. As a result, CALM iKD hardly affected the growth of HL-60, while it significantly impaired the growth of MV4-11 and HMC-1 as observed in Ba/F3 cells, indicating that CALM plays an important role in the growth AML cells carrying FLT3 ITD or KIT D816V. These results are shown as Fig. 1b.

(We have added these findings as Fig. 1b in Result sections as follows: Result section p.5 Line 126~ in the revised paper.)

Does depletion of CALM change the intracellular localization of mutated RTKs?

Response:

When CALM was knocked down with a DOX-inducible shRNA system in MV4-11 and HMC-1 cells, vesicle-like structures became to be undetectable at ER and endolysosomes, respectively, and both FLT3 ITD and KIT V814 spread in the cytoplasm

with a diffuse pattern in immunoelectron microscope analyses (shown as Fig. 4a and 4b).

Is the reduced signaling and/or proliferation due to CALM depletion further inhibited by CPZ in Ba/F3 cells or, more importantly, how do CALM-silenced AML cells respond to CPZ?

Response:

According to the suggestion, we examined the effects of CPZ on the growth of CALM KD Ba/F3-FLT3 ITD and Ba/F3-KIT D816V cells. In these cell lines, CPZ didn't further augment growth inhibitory effects of CALM KD in Ba/F3-FLT3 ITD and Ba/F3-KIT D814V cells (Fig.2b), suggesting that CPZ inhibits RT-MTK-dependent growth of Ba/F3 cells by inhibiting the function of CALM.

We also examined the effects of CPZ on the growth of AML cell lines MV4-11 and HMC-1, in which CALM was KD in an inducible manner. In contrast to CALM KD Ba/F3-FLT3 ITD and Ba/F3-KIT D814V cells, CPZ further suppressed the growth of CALM iKD MV4-11 and HMC-1 cells by about 15-20% (Fig. 3e).

Together, these results suggest that, although CPZ selectively targets AML cells with MT-RTKs mainly through CALM inhibition, inhibition of other molecules such as dopamine receptor (DR), serotone receptor (5-HTR) might, to some extent, contribute to full anti-AML activities of CPZ depending on cellular contexts. This speculation is supported by the fact that 5-HT and dopamine receptor antagonists Methiothepin (METHIO) and Thioridazine (THIO) partly inhibited the growth of MV4-11 and HMC-1 (Fig.3a). However, we also found that CPZ reduced CALM protein levels in MV4-11 and HMC-1 cells (Fig.3c). So, CPZ might further deplete residual CALM protein escaped from KD in these cells.

We have added these results in the Result and Discussion section as follows:

Result section p.6 Line 148~ in the revised paper (Fig. 2b).

Result section p.7 Line 191~ in the revised paper (Fig. 3e).

Discussion section p.13 Line 368~ in the revised paper.

Result section p.6 Line 174~ in the revised paper (Fig.3a).

Result section p.7 Line 185~ in the revised paper (Fig.3c.)

5. The authors do not discuss what the fate of FLT3 ITD receptor is when it is redistributed after CPZ from the ER. Does it accumulate in some Golgi subcompartment or elsewhere? Why do the two mutated receptors behave differently upon CPZ? These aspects, as well as the other above mentioned issues (after addressing them experimentally), should be interpreted in the discussion section which in the current version of the manuscript puts little focus on the molecular consequences of CPZ treatment.

Response:

After treatment with CPZ, both FLT3 ITD and KIT D816V spread in the cytoplasm with a diffuse pattern. Although the amount of FLT3 ITD was not influenced by CPZ treatment in MV4-11 cells (Fig. 5b), that of KIT D816V increased in HMC-1 cells after CPZ treatment from unknown mechanisms (Fig. 5d). To answer the reviewer's question, we performed immunoelectron microscope analyses. However, we couldn't determine the apparatus at which these RT-MTKs predominantly exist with this method. So, we further performed immunoprecipitation analyses using anti-Syntaxin (a marker for Golgi), anti-GM130 (for Golgi), and anti-EEA1 (for early endosomes) Abs. However, the localization patterns of these MT-RTKs didn't match those of any tested markers. Instead, we found that CPZ treatment decreased the expression of CALM protein in HMC-1 cells (Fig.3c). So, we assumed that intracellular trafficking of MT-RTKs and their subsequent location were totally impaired and disorganized by CPZ through downregulation of CALM protein.

(We have described these results in the result section as follows:

Result section from p.9 Line 243~ in the revised paper (Fig.5d).

Discussion section p.12 Line 352~ in the revised paper.

6. The text contains many wrong pieces of information about intracellular trafficking. For example: Introduction, the second sentence: “Receptor tyrosine kinase (RTK) is a high-affinity transmembrane protein, which is internalized by CME”. Not all RTKs are internalized by CME and in most known cases CME is only one of possible internalization routes.

Response:

We apologize for the wrong explanation about intracellular trafficking. As suggested by the reviewer, multiple different endocytic pathways have been described for RTKs. In

the revised version, we have deleted these sentences because we have focused on the localization of MT-RTKs but not on receptor endocytosis,

Introduction, line 64: “a part of CCVs containing WT RTKs is sorted back to the plasma membrane via recycling endosomes, while the remaining part is transported to late endosomes and consequently degraded at lysosomes”. CCVs are not sorted back via recycling endosomes or degraded at lysosomes. It is the membrane cargo that is sorted. Clathrin is removed from vesicles before fusing with endosomes and thus CCVs cease to exist once fused with endosomes.

Response:

We apologize for the wrong explanation about intracellular trafficking and degradation of CCV in Introduction.

We have corrected this sentence as follow: a part of WT-RTKs is sorted back to the plasma membrane via recycling endosomes (Introduction section from p 2 Line 48~ in the revised paper).

Discussion, the first sentence: “CALM functions in the initial step of CME by facilitating the formation of CCV, which is primarily found at the trans-Golgi network”. This statement is generally unclear but if the authors talk about CCVs mediating endocytosis, then these vesicles are formed at the plasma membrane, not at the TGN (where CCVs have other non-endocytic functions which are not discussed by the authors at all).

Response:

We appreciate a kind comment of the reviewer. We have deleted this sentence and the reference.

Overall, it is not clear why in the introduction and discussion the authors refer mainly to the effects of CPZ on endocytosis, while the phenotypes they observe in AML cells are more likely related to **other clathrin-mediated trafficking routes**, e.g. initiated at the TGN.

Response:

According to the reviewer's suggestion, we eliminated the part describing endocytosis in Introduction. Also, we modified introduction and discussion based on the results described in this manuscript as follows:

The structure of the N-terminal ANTH domain of CALM binds to phosphatidylinositol-4,5- biphosphate [PtdIns(4,5)P₂] and regulates the size and maturation of CCVs by recognizing membrane curvature²¹. Recent studies have shown that PtdIns(4,5)P₂ exists at multiple intracellular compartments, including the endosomes, lysosomes, ER, the Golgi complex, as well as at the plasma membrane³⁴. So, although the precise role of CALM protein in vesicle coat assembly has not been fully elucidated, it is plausible that CALM plays an important role in the formation of vesicles containing MT-RTKs at ER or endolysosomes and thereby regulates their transport. (Discussion section from p 12 Line 339~ in the revised paper).

Minor issues:

- The immunoblots showing changes in signaling should be quantified from several independent experiments (e.g. Fig. 1C, S2).

Response:

We quantified the changes of signaling in immunoblot analysis by several independent experiments.

- p. 3, line 69: The cited references (8 and 9) are outdated and do not include information about the mutated Kit receptor localization on the ER which should be cited (Obata et al. 2014 Nat Commun. 2014 Dec 10;5:5715. doi: 10.1038/ncomms6715).

Response:

We thank the reviewer for valuable information. We have included suggested reference in the revised manuscript.

- The description of Fig. 5 and 6 in the text is unclear. The authors do not explain which types of cells are CD45-positive or CD34-positive. They use interchangeably CD45 or LCA without explaining that it is the same marker.

Response:

We have indicated which types of cells are CD45-positive or CD34-positive. Also, as suggested by the reviewer, it is confusing to use both CD45 and LCA because CD45 and

LCA refer to the same cell surface marker. In the revised manuscript, we utilized only CD45 instead of LCA.

- Generally, the language (grammar/spelling) should be improved, below only a handful of more numerous examples:

- p. 3, line 71: missing “is” at the end of the line

- p. 3, line 79: missing “is” before “expressed”

- p. 4, line 90: missing “as” before “oncogene”

- p. 10-11, lines 288-289 Incorrect sentence. Should be: "... and even in cases of effective FLT3 inhibition, the effects were transient ..."

Response:

We thank the reviewer for pointing out our grammatical and spelling errors. We corrected them and checked the paper thoroughly to identify other mistakes.

- Table S1 should have some legend/explanation

Response:

We have added the Table legend in the Supplementary methods file.

- Fig. 4D Misplaced dashed line in #5?

Response: *We have corrected the position of the dashed line.*

Reviewers' comments:

Reviewer #2 (Remarks to the Author); expert in RTKs and vesicular trafficking:

Matsumura and colleagues addressed all the issues raised in my review, however not all the problems of the initial manuscript have been corrected. In addition, the revised version of the manuscript contains a number of new serious flaws which are unacceptable for any peer-reviewed journal.

1. The new title "Eradication of acute myeloid leukemia cells by altering the localization of mutant receptor tyrosine kinases through the inhibition of vesicle formation" implies that the authors have obtained evidence for inhibited vesicle formation in leukemia cells treated with CPZ or lacking CALM. Unfortunately, no such proof is presented as the authors essentially did not assess the distribution or abundance of any known markers of intracellular vesicles (endocytic or ER-derived). The nearest-to-be "vesicular" marker used in this study is clathrin heavy chain, whose abundance actually increases in CPZ-treated cells. However, the authors did not draw any conclusion from this observation. LAMP1 used in this study in some experiments is not a vesicular marker – it is a protein anchored to the limiting membranes of large endolysosomal structures (see also below).
The title should be changed.

2. There is a problematic inconsistency regarding the fate of the mutant KIT receptor upon CPZ treatment. According to Supplementary Fig. 4c, in MEFs KIT D814V is redistributed to fractions 7-25. However, these fractions, according to data regarding FLT3-ITD shown in the same panel, should represent PDI+ ER fractions. The authors still refrain from commenting on the identity of the compartment/s to which the mutated receptor is redistributed from the Golgi in CPZ-treated MEFs, although this issue was raised in the review. In HMC-1 cells, instead of analyzing where the mutated KIT is redistributed from the Golgi, the authors analyze its intracellular localization exclusively on structures which they call endolysosomes (see below, point 3). Why the authors do not analyze the presence of the mutated KIT in the ER/Golgi compartments of HMC-1 cells is unclear from the text. However, the literature data on the KIT receptor may help to understand this matter. The mutated KIT was found to signal from endosomes and from the ER in HMC-1 cells (Obata et al. Nat Commun. 2014). Conversely, in gastrointestinal stromal tumors (GIST), the mutant KIT accumulates on the Golgi apparatus (Obata et al. Oncogene, 2017). Thus, it seems that the results obtained by the authors in MEF cells represent the mechanism of receptor trafficking more similar to the one reported in GIST. The accumulation of the mutant KIT on the Golgi may likely not occur in HMC-1 suspension cells. Thus, as pointed out in my original review, it is very likely that membrane trafficking of receptors overexpressed in MEF cells is different than that of endogenously expressed in suspension cells.
The authors should remove entirely the supplementary Fig. 4 which has no importance for the delivered message and only provides confusion.

3. As requested, the authors analyzed the effect of CPZ or CALM depletion on the intracellular distribution of the mutated receptors in suspension cells, MV4-11 and HMC-1. Instead of performing immunofluorescence analysis as suggested, the authors applied immunoelectron microscopy (Fig. 4 and Supplementary Fig. 5). However, the quality of the obtained images is poor and not suitable for this type of analysis. In addition, the presented interpretation of these images is in some cases wrong. The authors identify white irregular holes in the cells as "endolysosomes". To confirm their identity, an immunostaining was performed which according to the authors showed LAMP1+ vesicle-like structures in the lumen of these "endolysosomes". Firstly, late endosomes and lysosomes are roundish organelles encircled by a clearly distinguishable limiting membrane and containing membranous intraluminal vesicles. No membranes (limiting or intraluminal) are visible on the presented micrographs. Secondly, LAMP1 protein, a common marker of late endosomes and lysosomes (hence endolysosomal structures), is anchored to the limiting membrane. The structures shown in the Supplementary Fig. 5 and Fig. 4 do not fulfill these two criteria and hence cannot be called "endolysosomes".

The authors continuously refrain from showing the IF analysis in suspension cells. The reason for this is unclear. HMC1 cells are suitable for immunofluorescence staining that can provide clear-cut and interpretable results, as shown for instance in Obata et al. Nat Commun. 2014.

↳ The authors should remove the inadequate immunoelectron microscopy images and replace them with conclusive results, preferably based on IF analysis.

4. The authors corrected wrong pieces of information in the previous manuscript. However, they introduced several false statements or incorrect conclusions regarding membrane trafficking, starting from the abstract, through the results and discussion sections, as well as in the graphical summary (Fig. 8).

a. Abstract, line 37: "...removed MT-RTKs from endoplasmic reticulum (ER) / endolysosomes" The authors provide no evidence that the MT-RTKs are removed from any compartment. The trafficking of these receptors may as well be disrupted in such a manner that the receptors never reach these compartments.

b. p. 8, line 235: "our result further suggests that FLT3 ITD must be present at ER in the form of CCVs to activate STAT5". This sentence is false in its essence. There are no CCVs at the ER. ER-derived vesicles are formed using other coating mechanisms which do not involve clathrin. Such a discovery would revolutionize the whole cell biology field but the authors have no data that would even remotely lead to this conclusion. The manuscript does not provide any information regarding the presence of clathrin chains on the ER. The obtained data points rather to an involvement of CALM in targeting/removal of FLT3 ITD to/from ER, which is not mechanistically proven in this manuscript, either.

c. The authors further refer to the above issue in the discussion section (p. 12 line 344): "it is plausible that CALM plays an important role in the formation of vesicles containing MT-RTKs at ER or endolysosomes and thereby regulates their transport". This concept is also presented in the graphical summary (Fig. 8). As the authors show no data regarding the intracellular localization of CALM with respect to the MT-RTKs and endosomal/ER markers, such concept should not be proposed. In the graphical summary the authors show that in untreated cells, vesicles harboring FLT3-ITD near or at the ER are positive for CALM, which is not proven in the manuscript. In turn, KIT D816V is shown as signaling from vesicles inside endolysosomes. It is not possible that receptors signal from the inside of endolysosomal structures. They need to be on the limiting membranes with their C-terminal ends facing the cytosol to activate signaling. Moreover, according to the depicted model, CPZ treatment leads to removal of both types of the mutated receptors from the CALM-positive vesicles and their dispersion in the cytoplasm. Given that these are transmembrane proteins their accumulation in the cytosol is very unlikely and not proven in the manuscript.

◇ The authors should remove the inappropriate sentences from the text and exchange the present graphical summary for the one which would reflect their actual findings and would be consistent with the cell biology text-book knowledge.

◇ Clearly, in order to draw firm conclusions in exchange for those that need to be removed, the authors need to study in more detail the identity of the intracellular compartments, from which and to which the receptors are relocated. This should include investigating whether CALM protein is present on any of these compartments (by IF or by IP).

Minor comment: Wrong information in line 232. "in the ER fraction" is likely a mistake - it should be "in the cell lysates".

Reviewer #3 (Remarks to the Author); expert in AML:

The manuscript by Matsumura and colleagues appears to be significantly improved and addresses all the main points raised previously by the Reviewer. Specifically, the mode of action of CPZ via modulation of intracellular localization of MT-RTKs is confirmed via both mechanistic studies (see figure 4 and 5 and related supplementary) and via the experiments with 5-HT and dopamine receptor antagonists.

The experiments where CALM KD was combined with CPZ further confirm the specificity of this mode of action in Ba/F3 cells and partially in human cell lines. However for the latter the authors discuss appropriately the reasons why CALM KD might have additive growth inhibitory effects with CPZ.

The experiments in primary samples and functionally defined LSC are much more convincing and the further work on the clinical case makes the paper much stronger. Overall I think this is an improved manuscript which has addressed all major criticism.

Minor comment:

the authors should clarify better in the text the findings shown in figure 5a and b. In figure 5a they show that in the ER fraction FLT3 ITD levels are reduced following CALM KD or CPZ treatment. However then they state that "the total amount of FLT3 ITD in the ER was not influenced by CPZ treatment" (page 8 line 232). They refer to figure 5b but this actually shows whole cell lysates rather than the ER fraction. So it is unclear to me how they come to the conclusion of page 8, line 232 to 236. This might need to be explained more clearly or, as I interpret it, the data show that total FLT3 ITD (but not in the ER fraction) does not change following CPZ treatment. Either way, this will not affect the overall conclusions of the paper and it is a minor point for revision.

Reviewer #2's comments

Matsumura and colleagues addressed all the issues raised in my review, however not all the problems of the initial manuscript have been corrected. In addition, the revised version of the manuscript contains a number of new serious flaws which are unacceptable for any peer-reviewed journal.

1. The new title “Eradication of acute myeloid leukemia cells by altering the localization of mutant receptor tyrosine kinases through the inhibition of vesicle formation” implies that the authors have obtained evidence for inhibited vesicle formation in leukemia cells treated with CPZ or lacking CALM. Unfortunately, no such proof is presented as the authors essentially did not assess the distribution or abundance of any known markers of intracellular vesicles (endocytic or ER-derived). The nearest-to-be “vesicular” marker used in this study is clathrin heavy chain, whose abundance actually increases in CPZ-treated cells. However, the authors did not draw any conclusion from this observation. LAMP1 used in this study in some experiments is not a vesicular marker – it is a protein anchored to the limiting membranes of large endolysosomal structures (see also below).

\ The title should be changed.

Response:

We appreciate this helpful suggestion. we have accordingly changed the title as follows:

Chlorpromazine reduces CALM protein and perturbs the intracellular localization of mutated receptor tyrosine kinases, thereby exhibiting anti-leukemic activities against acute myeloid leukemia cells with FLT3 ITD and KIT D816V

2. There is a problematic inconsistency regarding the fate of the mutant KIT receptor upon CPZ treatment. According to Supplementary Fig. 4c, in MEFs KIT D814V is redistributed to fractions 7-25. However, these fractions, according to data regarding FLT3-ITD shown in the same panel, should represent PDI+ ER fractions. The authors still refrain from commenting on the identity of the compartment/s to which the mutated receptor is redistributed from the Golgi in CPZ-treated MEFs, although this issue was raised in the review. In HMC-1 cells, instead of analyzing where the mutated KIT is redistributed from the Golgi, the authors analyze its intracellular localization exclusively on structures which they call endolysosomes (see below, point 3). Why the authors do not analyze the presence of the mutated KIT in the ER/Golgi

compartments of HMC-1 cells is unclear from the text. However, the literature data on the KIT receptor may help to understand this matter. The mutated KIT was found to signal from endosomes and from the ER in HMC-1 cells (Obata et al. Nat Commun. 2014). Conversely, in gastrointestinal stromal tumors (GIST), the mutant KIT accumulates on the Golgi apparatus (Obata et al. Oncogene, 2017). Thus, it seems that the results obtained by the authors in MEF cells represent the mechanism of receptor trafficking more similar to the one reported in GIST. The accumulation of the mutant KIT on the Golgi may likely not occur in HMC-1 suspension cells. Thus, as pointed out in my original review, it is very likely that membrane trafficking of receptors overexpressed in MEF cells is different than that of endogenously expressed in suspension cells.

\ The authors should remove entirely the supplementary Fig. 4 which has no importance for the delivered message and only provides confusion.

Response:

We thank the reviewer for pointing out this important observation. As suggested by the reviewer, the membrane trafficking of receptors overexpressed in MEF cells may be different from that of endogenously expressed in suspension cells. In line with these observations, in the revised manuscript we have eliminated former supplementary Fig. 4.

3. As requested, the authors analyzed the effect of CPZ or CALM depletion on the intracellular distribution of the mutated receptors in suspension cells, MV4-11 and HMC-1. Instead of performing immunofluorescence analysis as suggested, the authors applied immunoelectron microscopy (Fig. 4 and Supplementary Fig. 5). However, the quality of the obtained images is poor and not suitable for this type of analysis. In addition, the presented interpretation of these images is in some cases wrong. The authors identify white irregular holes in the cells as “endolysosomes”. To confirm their identity, an immunostaining was performed which according to the authors showed LAMP1+ vesicle-like structures in the lumen of these “endolysosomes”. Firstly, late endosomes and lysosomes are roundish organelles encircled by a clearly distinguishable limiting membrane and containing membranous intraluminal vesicles. No membranes (limiting or intraluminal) are visible on the presented micrographs. Secondly, LAMP1 protein, a common marker of late endosomes and lysosomes (hence endolysosomal structures), is anchored to the limiting membrane. The structures shown in the Supplementary Fig. 5 and Fig. 4 do not fulfill these two criteria and hence cannot be called “endolysosomes”. The authors continuously refrain from showing the IF analysis in suspension cells. The reason

for this is unclear. HMC1 cells are suitable for immunofluorescence staining that can provide clear-cut and interpretable results, as shown for instance in Obata et al. Nat Commun. 2014.

↳ The authors should remove the inadequate immunoelectron microscopy images and replace them with conclusive results, preferably based on IF analysis.

Response:

We thank the reviewer for these excellent suggestions. We have deleted the data obtained from immunoelectron microscopy. Instead, according to the suggestion of the reviewer, we have conducted immunofluorescence analyses using FLT3 ITD-positive MV4-11 cells and KITD816V-positive HMC1 cells. In this analysis, we utilized various compartment markers such as PDI (for ER), GM130 (for Golgi), EEA1 (for endosome), LAMP1 (for endolysosome), and Rab11 (for recycling endosome).

Our results indicate that CALM most abundantly colocalized with PDI (33.5% of total amount) and substantially colocalized with LAMP1 (20.8%) and EEA1 (15.3%) and partly with Rab11 (5.1%) and GM130 (3.1%) in MV4-11 cells. When CALM iKD MV4-11 cells were treated with DOX, the amount of CALM was effectively reduced by iKD. Also, in leukemia cells treated with CPZ, the expression of CALM protein was severely reduced.

Whereas in the absence of any treatment, nearly 80% of FLT3-ITD was co-localized with CALM, the proportion of FLT3 ITD co-localized with CALM was severely reduced from 79.1% to 4.2% by CALM iKD and to 6.9% by CPZ treatment due to CALM depletion.

In untreated cells, FLT3-ITD was co-localized with PDI (30.1%), with Rab11 (21.9%), and somewhat with GM130 (4.9%), EEA1 (8.6%), and LAMP1 (7.8%), suggesting that it is preferentially present at ER. However, CALM iKD or CPZ treatment reduced the co-localization of FLT3 ITD with PDI (to 7.1% by CALM iKD; to 6.1% by CPZ). In contrast, CALM iKD or CPZ treatment didn't influence the co-localization of FLT3 ITD with Rab11 (to 24.7% by CALM iKD; to 23.7% by CPZ), GM130 (to 4% by CALM iKD; to 4.2% by CPZ), EEA1 (to 6.7% by CALM iKD; to 7.4% by CPZ), or LAMP1 (to 6.6% by CALM iKD; to 6.6% by CPZ).

In untreated HMC-1 cells, CALM was co-localized with LAMP1 (20.8%), EEA1 (15.2%), PDI (12.9%), GM130 (4.3%), and Rab11 (1.9%). As observed in MV4-11 cells, CALM iKD or CPZ treatment apparently reduced CALM protein. Although 86.5% of KIT D816V was co-localized with CALM, this co-localization was impaired by CALM iKD or CPZ treatment due to CALM depletion (to 7.1% by CALM iKD; to 12.2% by CPZ).

We next investigated the intracellular localization of KIT D816V in HMC-1 cells. Without any treatment, KIT D816V was co-localized with LAMP1 (26.4%), with EEA1 (17%), and somewhat

with PDI (13.7%), GM130 (10.6%), and Rab11 (4.8%), suggesting that it is predominantly present in endolysosome. In contrast, CALM iKD or CPZ treatment apparently reduced co-localization of KIT D816V with LAMP1 (to 7.2% by CALM iKD; to 8% by CPZ), with EEAI (to 6.3% by CALM iKD; to 6.1% by CPZ), and with PDI (to 8.4% by CALM iKD; to 8.3% by CPZ), GM130 (to 7.1% by CALM iKD; to 5.9% by CPZ) excepting for the co-localization with Rab11 (to 4.6% by CALM iKD; to 4.3% by CPZ).

In accord with these results, we have modified the figures and text:

Result section from p.7 Line 200 to p.9 Line252 in the revised paper (figure 4 and supplementary figure4 in the revised version)

Because neither FLT ITD nor KIT D816V apparently co-localized with any marker, we couldn't determine at which compartment they were present after CALM iKD or CPZ treatment. So, we have described this issue as follows (Result section from p.9 Line251 to 252 in the revised paper):

Together, these results indicate that CPZ reduces CALM protein and perturbs the intracellular localization of FLT ITD and KIT D816V, thereby inhibiting their oncogenic signals.

4. The authors corrected wrong pieces of information in the previous manuscript. However, they introduced several false statements or incorrect conclusions regarding membrane trafficking, starting from the abstract, through the results and discussion sections, as well as in the graphical summary (Fig. 8).

a. Abstract, line 37: "...removed MT-RTKs from endoplasmic reticulum (ER) / endolysosomes" The authors provide no evidence that the MT-RTKs are removed from any compartment. The trafficking of these receptors may as well be disrupted in such a manner that the receptors never reach these compartments.

Response:

We thank the reviewer for pointing out this important observation. We have corrected the description in the abstract as follows (Abstract section from p.2 Line 36 to 37 in the revised paper):

Mechanistically, CPZ reduces CALM protein at post transcriptional level and perturbs the intracellular localization of MT-RTKs, thereby blocking their signaling.

Also, we have deleted Fig. 8 of the previous paper.

b. p. 8, line 235: “our result further suggests that FLT3 ITD must be present at ER in the form of CCVs to activate STAT5”. This sentence is false in its essence. There are no CCVs at the ER. ER-derived vesicles are formed using other coating mechanisms which do not involve clathrin. Such a discovery would revolutionize the whole cell biology field but the authors have no data that would even remotely lead to this conclusion. The manuscript does not provide any information regarding the presence of clathrin chains on the ER. The obtained data points rather to an involvement of CALM in targeting/removal of FLT3 ITD to/from ER, which is not mechanistically proven in this manuscript, either.

Response:

We would like to thank the reviewer for the kind criticism. We have deleted this sentence.

c. The authors further refer to the above issue in the discussion section (p. 12 line 344): “it is plausible that CALM plays an important role in the formation of vesicles containing MT-RTKs at ER or endolysosomes and thereby regulates their transport”. This concept is also presented in the graphical summary (Fig. 8). As the authors show no data regarding the intracellular localization of CALM with respect to the MT-RTKs and endosomal/ER markers, such concept should not be proposed. In the graphical summary the authors show that in untreated cells, vesicles harboring FLT3-ITD near or at the ER are positive for CALM, which is not proven in the manuscript. In turn, KIT D816V is shown as signaling from vesicles inside endolysosomes. It is not possible that receptors signal from the inside of endolysosomal structures. They need to be on the limiting membranes with their C-terminal ends facing the cytosol to activate signaling. Moreover, according to the depicted model, CPZ treatment leads to removal of both types of the mutated receptors from the CALM-positive vesicles and their dispersion in the cytoplasm. Given that these are transmembrane proteins their accumulation in the cytosol is very unlikely and not proven in the manuscript.

◇ The authors should remove the inappropriate sentences from the text and exchange the present graphical summary for the one which would reflect their actual findings and would be consistent with the cell biology text-book knowledge.

◇ Clearly, in order to draw firm conclusions in exchange for those that need to be removed, the authors need to study in more detail the identity of the intracellular compartments, from which and to which the receptors are relocated. This should include investigating whether CALM protein is present on any of these compartments (by IF or by IP).

Response:

We thank the reviewer for pointing out these important observations. Accordingly, we have deleted the inappropriate description and discussion. Also, in the revised manuscript we have removed former Fig. 8.

Minor comment: Wrong information in line 232. “in the ER fraction” is likely a mistake - it should be “in the cell lysates”.

Response:

*We have corrected from “in the ER fraction” to “**in the whole cellular lysates**” (page8. line 226 in the revised paper).*

Reviewer #3 (Remarks to the Author)

The manuscript by Matsumura and colleagues appears to be significantly improved and addresses all the main points raised previously by the Reviewer. Specifically, the mode of action of CPZ via modulation of intracellular localization of MT-RTKs is confirmed via both mechanistic studies (see figure 4 and 5 and related supplementary) and via the experiments with 5-HT and dopamine receptor antagonists.

The experiments where CALM KD was combined with CPZ further confirm the specificity of this mode of action in Ba/F3 cells and partially in human cell lines. However for the latter the authors discuss appropriately the reasons why CALM KD might have additive growth inhibitory effects with CPZ.

The experiments in primary samples and functionally defined LSC are much more convincing and the further work on the clinical case makes the paper much stronger. Overall I think this is an improved manuscript which has addressed all major criticism.

Minor comment:

the authors should clarify better in the text the findings shown in figure 5a and b. In figure 5a they show that in the ER fraction FLT3 ITD levels are reduced following CALM KD or CPZ treatment. However then they state that "the total amount of FLT3 ITD in the ER was not influenced by CPZ treatment" (page 8 line 232). They refer to figure 5b but this actually shows whole cell lysates rather than the ER fraction. So it is unclear to me how they come to the conclusion of page 8, line 232 to 236. This might need to be explained more clearly or, as I interpret it, the data show that total FLT3 ITD (but not in the ER fraction) does not change following CPZ treatment. Either way, this will not affect the overall conclusions of the paper and it is a minor point for revision.

Response:

The reviewer is right and we would like to apologize for the unclear statements about Flt3 protein shown in figure 5a and b of the previous manuscript. As reviewer mentioned, it was

incorrect to rely that total FLT3 ITD, but not that in the ER fraction did not change following CPZ treatment. In the revised manuscript, we have changed the phrase “in the ER fraction” to “in the whole cellular lysates” (page8. line 226 in the revised paper).

REVIEWERS' COMMENTS:

Reviewer #2 (Remarks to the Author):

The revised manuscript is overall consistent and the results obtained justify the conclusions drawn. The authors removed controversial pieces of data and inappropriate statements and performed the immunofluorescence (IF) analysis that was missing from the previous version. This reviewer is only concerned that the quantification of IF analysis was performed on a low number of cells (only 10 cells from which SEM was calculated). However, the obtained IF results are consistent with the other findings, therefore are likely true. The manuscript will be suitable for publication after addressing one major and some minor comments.

Major comment:

1) The authors tested redistributions of the mutated receptors with respect to markers of several intracellular compartments. They concluded that the receptors undergo redistribution from various compartments because their colocalizations with most markers were reduced. However, it remains unknown to which cellular compartment(s) the mutated receptors are redistributed, likely because some important markers have not been tested. CALM is a clathrin adaptor, therefore perturbing its function could lead to accumulation of the mutated receptors on clathrin-positive structures, that originate at the PM or at late Golgi membranes. Especially the reduced receptor localization at endolysosomes could result from inhibition of clathrin-mediated trafficking from the PM or the Golgi. The presented images do not suggest PM accumulation, however the Golgi marker tested by the authors, GM130, is a cis-Golgi marker and does not represent the structures involved in Golgi-to-lysosome trafficking.

This knowledge, although very important, would not change the final conclusions of the paper, therefore the authors are not required to perform further experiments. However, such interpretation is missing from the discussion section and should be therefore briefly provided.

Minor comments:

- 1) The information is missing regarding the IF images presented in Fig. 4 and S4. Are they single confocal images across the middle z-stack of the cells, z-stack projection images, or else?
- 2) The y axis label in Fig. 4f contains "FKIT". Likely a mistake and should be "KIT".
- 3) line 239: "predominantly present" should be exchanged for "preferentially present" (as in line 219) or better, "enriched". The signal is the highest among the analyzed compartments, but not necessarily throughout the whole cell. 26% Manders colocalization does not mean predominant localization (it is a quarter of the whole signal).
- 4) line 242: "excepting for" - grammatically incorrect
- 5) line 354: "intracellular trafficking of MT-RTKs and their subsequent localization of MT-RTKs" - 2nd "of MT-RTKs" should be removed.
- 6) line 389: " studies reported CPZ revealed anti-tumor activities against colorectal cancer cell lines". Incorrect sentence. Maybe "revealed" should be removed.

REVIEWERS' COMMENTS:

Reviewer #2 (Remarks to the Author):

The revised manuscript is overall consistent and the results obtained justify the conclusions drawn. The authors removed controversial pieces of data and inappropriate statements and performed the immunofluorescence (IF) analysis that was missing from the previous version. This reviewer is only concerned that the quantification of IF analysis was performed on a low number of cells (only 10 cells from which SEM was calculated). However, the obtained IF results are consistent with the other findings, therefore are likely true. The manuscript will be suitable for publication after addressing one major and some minor comments.

Major comment:

1) The authors tested redistributions of the mutated receptors with respect to markers of several intracellular compartments. They concluded that the receptors undergo redistribution from various compartments because their colocalizations with most markers were reduced. However, it remains unknown to which cellular compartment(s) the mutated receptors are redistributed, likely because some important markers have not been tested. CALM is a clathrin adaptor, therefore perturbing its function could lead to accumulation of the mutated receptors on clathrin-positive structures, that originate at the PM or at late Golgi membranes. Especially the reduced receptor localization at endolysosomes could result from inhibition of clathrin-mediated trafficking from the PM or the Golgi. The presented images do not suggest PM accumulation, however the Golgi marker tested by the authors, GM130, is a cis-Golgi marker and does not represent the structures involved in Golgi-to-lysosome trafficking.

This knowledge, although very important, would not change the final conclusions of the paper, therefore the authors are not required to perform further experiments. However, such interpretation is missing from the discussion section and should be therefore briefly provided.

Response:

We thank the reviewer for pointing out this very important observation. As suggested by the reviewer, we have provided the interpretation in the discussion section as follows:

Because CALM is a clathrin adaptor, we speculated that perturbing its function by CALM iKD or CPZ could lead to accumulation of the mutated receptors on clathrin-positive structures, that originate

at the plasma membrane (PM) or at late Golgi membranes. Furthermore, the reduced receptor localization at endolysosomes could result from inhibition of clathrin-mediated trafficking from the PM or the Golgi. However, we couldn't determine to which cellular compartments the mutated receptors are redistributed due to the limitation of the utilized markers. The presented images do not suggest PM accumulation, however GM130 is a cis-Golgi marker and does not represent all structures involved in Golgi-to-lysosome trafficking. So, further studies are needed to answer this issue.

Minor comments:

1) The information is missing regarding the IF images presented in Fig. 4 and S4. Are they single confocal images across the middle z-stack of the cells, z-stack projection images, or else?

Response:

We have provided the information regarding the IF images presented in Fig. 4 and S4 in Material and Methods as follows:

Immunofluorescence analysis.

Each image showed single sections with an x60 oil immersion objective, adjusted to give the same x, y, and z position in the all channels.

2) The y axis label in Fig. 4f contains "FKIT". Likely a mistake and should be "KIT".

Response:

We have corrected from "FKIT" to "KIT" in Fig. 4f.

3) line 239: "predominantly present" should be exchanged for "preferentially present" (as in line 219) or better, "enriched". The signal is the highest among the analyzed compartments, but not necessarily throughout the whole cell. 26% Manders colocalization does not mean predominant localization (it is a quarter of the whole signal).

Response:

We thank the reviewer for pointing out the important observations. Accordingly, we have corrected from "predominantly present" to "enriched". (page8. line 233 in the revised paper).

4) line 242: "excepting for" - grammatically incorrect

Response:

We have corrected from “excepting for” to “except”. (page8. line 236 in the revised paper).

5) line 354: "intracellular trafficking of MT-RTKs and their subsequent localization of MT-RTKs" - 2nd "of MT-RTKs" should be removed.

Response:

We have deleted the 2nd "of MT-RTKs". (page8. line 236 in the revised paper).

6) line 389: " studies reported CPZ revealed anti-tumor activities against colorectal cancer cell lines". Incorrect sentence. Maybe "revealed" should be removed.

Response:

We have corrected from “revealed” to “has”. (page8. line 236 in the revised paper).